# Fine-tuning mechanical constraints reveals uncoupled patterning and gene expression programs in murine gastruloids

Judith Pineau[1,*], Jerome Wong-Ng[2,*], Alexandre Mayran[3], Lucille Lopez-Delisle[3], Pierre Osteil[3], Armin Shoushtarizadeh[1], Denis Duboule[3,4,‡], Samy Gobaa[2,‡] and Thomas Gregor[1,5,‡,§]

## ABSTRACT

The interplay between mechanical forces and genetic programs is fundamental to embryonic development, yet how these factors influence morphogenesis and cell fate decisions remains unclear. Here, we fine-tune the mechanical environment of murine gastruloids, three-dimensional *in vitro* models of early embryogenesis, by embedding them in bioinert hydrogels with tunable stiffness and timing. This strategy reveals that external constraints can selectively influence transcriptional profiles, patterning or morphology, depending on the level and timing of mechanical modulation. Gastruloids in ultra-soft hydrogels (<30 Pa) elongate robustly, preserving anteroposterior patterning and transcriptional profiles. In contrast, embedding at higher stiffness disrupts polarization while leaving gene expression largely unaffected. Conversely, earlier embedding significantly impacts transcriptional profiles independently of polarization defects, highlighting the uncoupling of patterning and transcription. These findings suggest that distinct cellular states respond differently to external constraints. Live imaging and cell tracking further suggest that impaired cell motility underlies polarization defects, underscoring the role of mechanical forces in shaping morphogenesis independently of transcriptional changes. By precisely controlling mechanical boundaries, our approach provides a powerful platform to dissect how physical and biochemical factors interact to orchestrate embryonic development.

KEY WORDS: Pseudoembryos, Synthetic extracellular matrix, Polarization, Morphogenesis

## INTRODUCTION

Despite significant advances in developmental biology, the mechanisms that drive early embryogenesis – symmetry breaking, axis formation, germ layer specification and tissue morphogenesis –

remain incompletely understood. These processes arise from a complex interplay of genetic, biochemical and mechanical signals (Collinet and Lecuit, 2021), yet their precise interactions and temporal coordination remain elusive. Mammalian *in vivo* models are particularly challenging for such studies, as they are highly sensitive and constrained by the fact that these events occur post-implantation, making it difficult to disentangle the respective contributions of mechanical forces and biochemical cues. These limitations underscore the importance of controlled *in vitro* systems to systematically explore the role of physical and molecular factors in early development.

Recent advances in stem cell biology have enabled the creation of three-dimensional models known as gastruloids, which recapitulate key events of early mammalian embryogenesis. Gastruloids self-organize into aggregates that mimic symmetry breaking, anteroposterior (AP) axis elongation, and germ layer specification, providing a powerful platform for studying early developmental processes (van den Brink et al., 2014; Turner, et al., 2017; Beccari et al., 2018). Embedding these structures in extracellular matrix (ECM) substitutes, such as Matrigel, has demonstrated the crucial role of mechanical properties in morphogenesis and cell fate determination (Veenvliet et al., 2020; van den Brink et al., 2020; Hamazaki et al., 2024; Muncie et al., 2020). However, the undefined chemical composition of Matrigel and its inherent mechanical properties are inextricably linked, making it impossible to separate these effects. Additionally, batch-to-batch variability poses significant challenges for quantitative studies (Hughes et al, 2010; Vukicevic et al., 1992). Overcoming these limitations requires new approaches that combine controlled mechanical environments with high-resolution imaging to uncover how physical forces and biochemical signals coordinate development.

In this study, we leverage bioinert hydrogels with tunable stiffness to precisely control the mechanical environment of murine gastruloids. By systematically modulating both the stiffness and timing of embedding, we uncover how external constraints selectively influence transcriptional profiles, AP patterning, and morphology. Gastruloids embedded in ultra-soft hydrogels (<30 Pa) elongate robustly while preserving both transcriptional profiles and AP patterning, mimicking the behavior of controls. In contrast, stiffer hydrogels (>30 Pa) can disrupt polarization without altering gene expression, whereas earlier embedding significantly impacts transcriptional profiles independently of polarization defects. These findings reveal a surprising decoupling of transcriptional programs and AP patterning under specific mechanical conditions, challenging the conventional view of their tight coordination.

In addition to uncovering the uncoupling of transcriptional programs and patterning, our system minimizes sample movement during live imaging, enabling precise tracking of cell motility and

[1]Department of Developmental and Stem Cell Biology, CNRS UMR3738 Paris Cité, Institut Pasteur, 75015 Paris, France. [2]Biomaterials and Microfluidics Core Facility, Université Paris Cité, Institut Pasteur, 75015 Paris, France. [3]Swiss Institute for Experimental Cancer Research - ISREC, School of Life Sciences, Ecole Polytechnique Fédérale de Lausanne (EPFL), 1015 Lausanne, Switzerland. [4]Center for Interdisciplinary Research in Biology (CIRB), Collège de France, CNRS, UMR 7241, INSERM, Université PSL, 75005 Paris, France. [5]Joseph Henry Laboratories of Physics & Lewis-Sigler Institute for Integrative Genomics, Princeton University, Princeton, NJ 08544, USA.
*These authors contributed equally to this work
‡These authors contributed equally to this work

§Author for correspondence (thomas.gregor@pasteur.fr)

J.P., 0000-0003-0665-1210; J.W.-N., 0000-0001-8287-5203; A.M., 0000-0002-1228-0308; L.L.-D., 0000-0002-1964-4960; P.O., 0000-0002-5832-6703; D.D., 0000-0001-9961-2960; T.G., 0000-0001-9460-139X

morphogenesis. This advancement allowed us to identify impaired cell motility as a contributing factor underlying polarization defects in stiffer hydrogels. By providing precise control over mechanical constraints, our approach reveals how finely tuned environments can selectively influence distinct developmental outcomes. Together, these findings establish embedded gastruloids as a robust and versatile platform for probing the interplay between genetic, biochemical and physical factors in early embryogenesis. By shaping our understanding of how mechanical environments guide developmental processes, this work offers insights into the regulatory principles of embryogenesis.

## RESULTS
### Embedding in ultra-soft hydrogels allows reproducible gastruloid elongation
We developed an embedding procedure using a dextran-based hydrogel to investigate gastruloid elongation in a mechanically and chemically controlled environment (Fig. 1A,C). By varying hydrogel concentrations from 0.7 mM to 1.5 mM, we achieved stiffnesses ranging from 1 to 300 Pa (Fig. 1B, Fig. S1A), encompassing the range reported to support gastruloid elongation (Veenvliet et al., 2020; van den Brink et al., 2020). Importantly, this bioinert hydrogel minimizes extraneous signaling and variability, contrasting with traditional matrices such as Matrigel (Aisenbrey and Murphy, 2020; Blache et al., 2022) (details in Materials and Methods and Fig. 1).

We then compared the morphology of gastruloids embedded in hydrogels to those grown under standard culture conditions (Ctrl, no hydrogel). Gastruloids were prepared from 129/svev mouse embryonic stem cells (mESCs) cultured in Serum+2i+LIF conditions, which allows a homogeneous starting cell population and therefore robust organoid formation (Merle et al., 2023 preprint). Embedding was performed at 96 h post-seeding and gastruloids were analyzed at 120 h, the time frame during which the AP axis typically develops in our culture conditions.

Gastruloids embedded in hydrogels with concentrations below 1.0 mM successfully elongated, achieving approximately 80% of the medial axis length observed in controls (Fig. 1D-F, Fig. S1B-F). Quantification of the elongation index also showed a strong, robust decrease in elongation upon embedding in 1.0 mM gels, but very limited, variable effects of embedding at lower concentrations (Fig. 1F, Fig. S1D,F). Interestingly, these embedded gastruloids exhibited a straighter morphology, as quantified by an increased straightness ratio, compared to controls (Fig. 1D,G, Fig. S1B-F). This suggests that while the mechanical constraints of the hydrogel did not prevent elongation, they counterbalanced bending forces, promoting straighter contours during elongation. Such straighter contours reduce shape variability, a key advantage for quantitative analyses and for interpreting morphological measurements.

In contrast, gastruloids embedded in higher stiffness hydrogels (1.0 mM) showed limited to no elongation (Fig. 1E,F, Fig. S1D,F), and their straightness ratio approached 1 (Fig. 1G, Fig. S1D,F), indicative of a lack of significant morphological changes. These findings demonstrate that the mechanical properties of the environment directly influence the elongation process, with ultrasoft hydrogels (<1.0 mM) providing sufficient support for robust and reproducible elongation, while higher stiffness disrupts this process.

The embedding process also offers unique advantages for imaging and quantitative assays. By stabilizing gastruloids within a mechanically stable hydrogel, thermal fluctuations that often interfere with live imaging are minimized, enabling precise tracking of gastruloid dynamics. Additionally, reduced morphological

variability, as indicated by straighter contours, facilitates reproducible quantitative measurements. Finally, hydrogel embedding provides a means to separate mechanical and chemical contributions to gastruloid development, highlighting its potential as an alternative to traditional assays employing Matrigel.

Interestingly, we also observed that embedding gastruloids in dextran-based, bioinert gels produced markedly different outcomes compared to Matrigel (Fig. S2). Specifically, embedding in an ultrasoft, bioinert gel at 96 h post-seeding permitted uniaxial elongation by 120 h but did not enhance organoid maintenance at later time points. In contrast, gastruloids embedded in Matrigel at the same stage developed pronounced multipolar morphologies by 120 h but exhibited sustained, non-collapsing elongation by 144 h post-seeding. This difference is unlikely to result solely from signaling molecules or nutrients in Matrigel, as supplementing the media with Matrigel on top of embedded gastruloids failed to reproduce the phenotype (Fig. S2). Instead, the observed behavior likely stems from the ability of cells to adhere to, remodel and degrade Matrigel. Supporting this, RNA sequencing (RNA-seq) revealed expression of metalloproteases in our system (Fig. S5E), suggesting that the mechanical properties of Matrigel are altered over time through cellular remodeling, in contrast to the constant mechanical landscape offered by bioinert hydrogels.

Together, these results establish a robust platform for studying gastruloid development in controlled mechanical environments, providing both physiological relevance and improved reproducibility for imaging and quantitative workflows.

### Mechanical embedding preserves gastruloid patterning and transcriptional profiles
During gastruloid development, the establishment of the AP axis is closely linked to axis elongation and patterned expression of key germ layer markers, such as brachyury (BRA; T) and SOX2 (posterior end) or FOXC1 (anterior end) (Blassberg et al., 2022; Turner et al., 2014; van den Brink et al., 2014; Mittnenzweig et al., 2021). To assess whether this patterning is maintained in hydrogels, we performed immunofluorescence staining and quantified intensity profiles along the AP axis (Merle et al., 2023 preprint).

Remarkably, a BRA/SOX2 pole was observed under all conditions, even in gastruloids grown in higher-stiffness hydrogels (1.0 mM), where elongation was impaired (Fig. 2A, Fig. S3A,C). To account for differences in fixation protocols between embedded and non-embedded samples (see Materials and Methods), fluorescence intensities were normalized both for signal range and for medial axis length (Merle et al., 2023 preprint). Specifically, intensity values were scaled for each experiment relative to the average profile of all gastruloids, using the 10% lowest and highest values for reference. Spatial profiles were normalized to the medial axis length of individual gastruloids. Gastruloids embedded in ultra-soft gels (0.7-0.8 mM) exhibited normalized SOX2 and BRA expression profiles that closely matched those of non-embedded controls (Ctrl, no hydrogel) grown in standard culture conditions, as exhibited by the lack of consistent differences in the boundary positions of these markers between these different conditions (Fig. 2A-C, Fig. S3). By contrast, samples embedded in 1.0 mM gels showed spatial deviations, likely reflecting limited elongation and reduced consistency in AP axis alignment during imaging (Fig. 2A-C, Fig. S3). Importantly, embedding in ultra-soft hydrogel at 96 h post-seeding did not significantly alter FOXC1 patterned expression at the anterior end either (Fig. S4), further supporting that ultra-soft hydrogels preserve AP patterning comparable to controls.

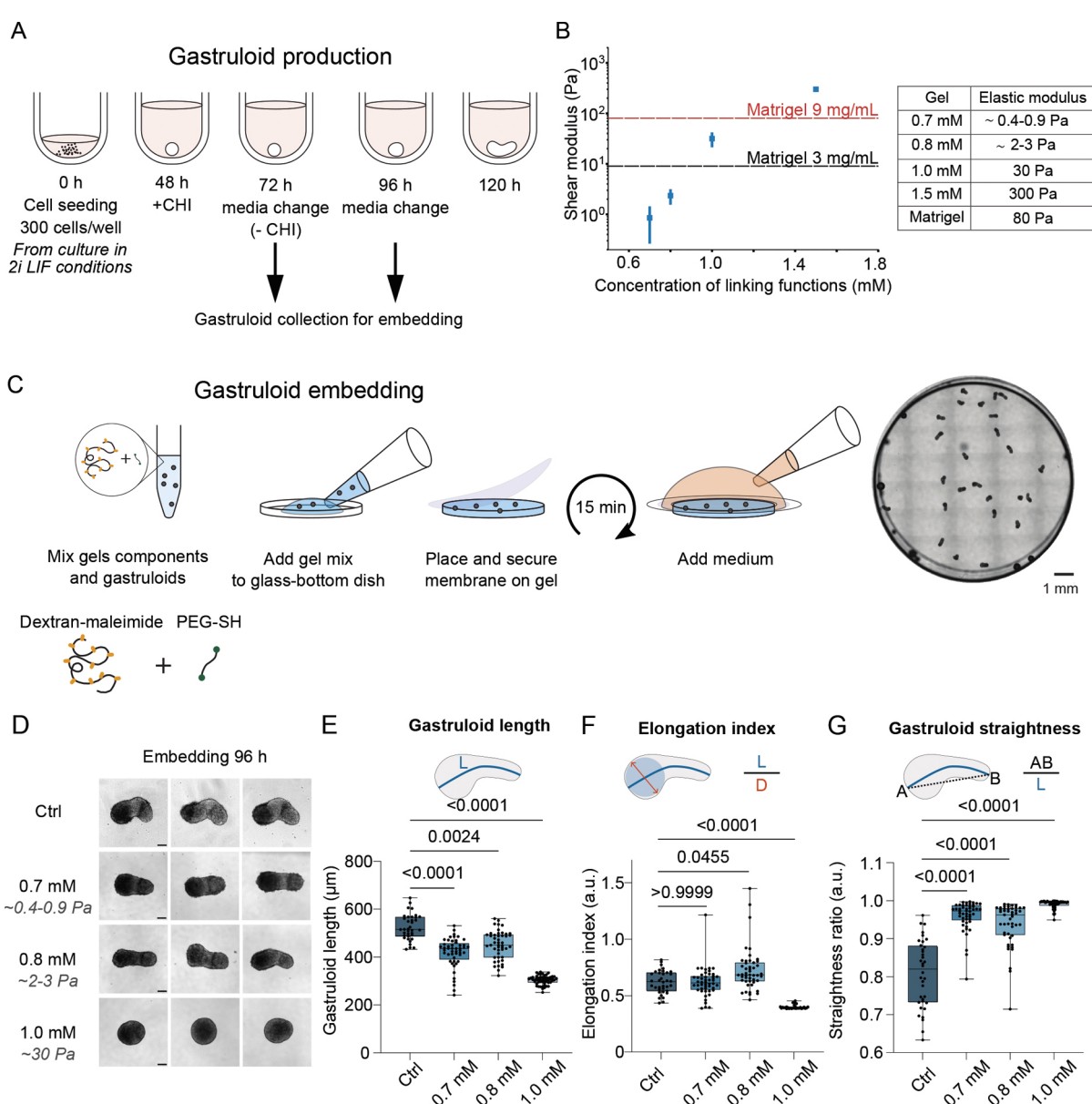

**Fig. 1. Embedding gastruloids in an ultra-low stiffness dextran-based hydrogel.** (A) Schematic view of the gastruloid generation protocol. (B) Left: Stiffness of hydrogels prepared with different concentrations of components (concentration being the concentration of reactive functions), measured using a rheometer. mean±s.d. (n=3). Right: Table with elastic moduli of hydrogels as a function of concentrations of reactive functions. (C) Left: Protocol of gastruloid embedding, using a mix of Dextran-Maleimide and a polyethylene-glycol-thiol linker (PEG-SH). Right: Overview of a dish of embedded gastruloids at 120 h. (D) Representative brightfield images of gastruloids 120 h after seeding. Gastruloids were grown in standard culture conditions (Ctrl, no hydrogel) or embedded in hydrogels with increasing concentrations (0.7 mM, 0.8 mM or 1.0 mM) at 96 h post-seeding. Scale bars: 100 µm. (E) Quantification of gastruloid medial axis length (L) at 120 h, showing moderately reduced elongation with increasing hydrogel stiffness. Data correspond to the conditions shown in D. Sample sizes: Ctrl, n=36; 0.7 mM, n=50; 0.8 mM, n=47; 1.0 mM, n=56. (F) Elongation index of the same gastruloids as in E, showing very little effect of embedding in low-stiffness hydrogels, and strong limitation of elongation in high-stiffness hydrogels (1.0 mM). (G) Straightness ratio of the same gastruloids as in E, revealing increased morphological straightness when embedded in hydrogel. Straightness is calculated as illustrated in the schematic. Statistical tests were performed using the Kruskal–Wallis test with Dunn's multiple comparison test. Shown here are representative data and analysis for one experimental replicate. In E-G, box limits represent first and third quartiles, whiskers minimum and maximum values and horizontal line the median. See Fig. S1 for two additional replicates.

To evaluate whether gel embedding affects gene expression more broadly, we performed bulk RNA-seq. As a negative control, we included gastruloids that did not receive a CHIR99021 pulse (noCHI), a condition under which gastruloids remain spherical and fail to establish germ layers (van den Brink et al., 2014). Principal component analysis (PCA) on the top 500 most variable genes showed that 50% of the variance was explained by the difference between noCHI and all other conditions, while only 13% of the variance was attributed to differences among embedded and control gastruloids (Fig. 2D, Fig. S5A,B). Gastruloids embedded in ultra-soft hydrogels clustered closely with controls and no clear spatial separation was observed in the PCA, indicating high similarity between embedded and control conditions (Fig. 2D).

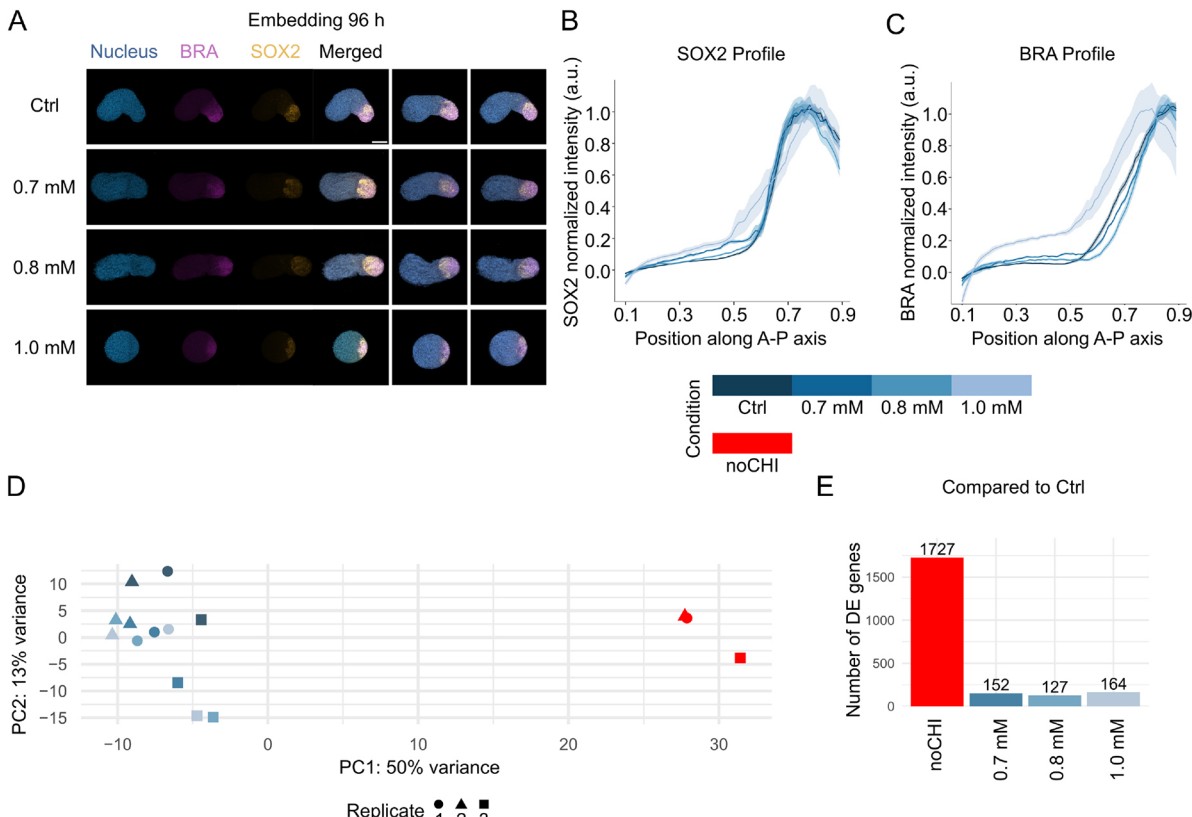

**Fig. 2. Gene expression patterning and transcriptional profiles in hydrogel-grown gastruloids.** (A) Representative immunofluorescence images of gastruloids at 120 h. Gastruloids were grown in standard culture conditions (Ctrl, no hydrogel) or embedded in hydrogels with increasing concentrations (0.7 mM, 0.8 mM and 1.0 mM) at 96 h post-seeding. Nuclei (blue), BRA (purple) and SOX2 (yellow) are labeled. Scale bar: 100 μm. (B,C) Normalized expression profiles (mean±s.e.m.) of SOX2 (B) and BRA (C) along the AP axis, demonstrating consistent expression patterns across conditions. Sample sizes: Ctrl, *n*=18; 0.7 mM, *n*=6; 0.8 mM, *n*=19; 1.0 mM, *n*=10. (D) Principal component analysis (PCA) of bulk RNA-seq experiments (three replicates), revealing clear separation between Ctrl and noCHI (red), a negative control for gastruloid formation, while hydrogel-embedded conditions cluster near Ctrl. (E) Number of significantly differentially expressed (DE) genes compared to Ctrl, showing substantial transcriptional changes in noCHI and minimal changes in hydrogel-embedded conditions (0.7 mM, 0.8 mM and 1.0 mM). Shown here are representative data and analysis for one experimental replicate. See Fig. S3 for two additional replicates.

The number of significantly differentially expressed (DE) genes between embedded and control gastruloids was small (Fig. 2E, Table S1). Furthermore, we found no effect of gel concentration on transcriptional profiles (Fig. 2D), suggesting that preventing elongation in higher-stiffness hydrogels (1.0 mM) does not strongly impact gene regulation. About half of DE genes were shared between all three gel concentrations (Fig. S5C). This suggests that these differences may arise from the addition of the external boundary condition or the de-embedding process required for RNA-seq sample preparation.

To assess further whether embedding influences the cellular composition of gastruloids, we deconvolved our bulk RNA-seq data using a previously published single-cell RNA-seq atlas of gastruloid development (Mayran et al., 2023 preprint) (Fig. S5D). This analysis revealed substantial batch-to-batch variability, with differences between embedded and non-embedded samples being less marked than those observed across batches and lacking consistent trends. Moreover, no systematic effect of hydrogel concentration on inferred cell type proportions was detected. These results suggest that neither the embedding procedure nor the stiffness of the hydrogel significantly alters the cellular composition of gastruloids. This supports the use of hydrogel-embedded gastruloids as a valid assay for probing developmental processes without introducing compositional artifacts.

Altogether, these results demonstrate that embedding gastruloids in ultra-soft hydrogels preserves AP patterning, transcriptional profiles and cellular composition, making this approach a viable alternative to traditional ECM-based matrices like Matrigel, which often exhibit batch-to-batch variability and ill-defined chemical compositions (Hughes et al., 2010; Vukicevic et al., 1992). The minimal transcriptional and morphological deviations observed suggest that gastruloids grown in these conditions can be used interchangeably with those grown in standard culture, enabling the study of gastrulation in a mechanically controlled environment.

### Timing and stiffness reveal uncoupling of patterning and gene expression

As patterning and transcriptional profiles appeared robust to changes in the mechanical environment, we next sought to determine the limits of their establishment. Specifically, we tested whether increasing environmental stiffness to ~300 Pa (Fig. 1B, Figs S1A, S6) or applying mechanical constraints earlier in development (Figs S6, S7) could disrupt these processes.

To assess the impact of stiffness on patterning, we first examined whether gastruloids could organize a singular BRA/SOX2 pole at 120 h post-seeding when embedded in stiffer hydrogels at 96 h. Remarkably, most gastruloids formed BRA/SOX2 poles at 120 h, regardless of gel stiffness, with approximately 75% showing a

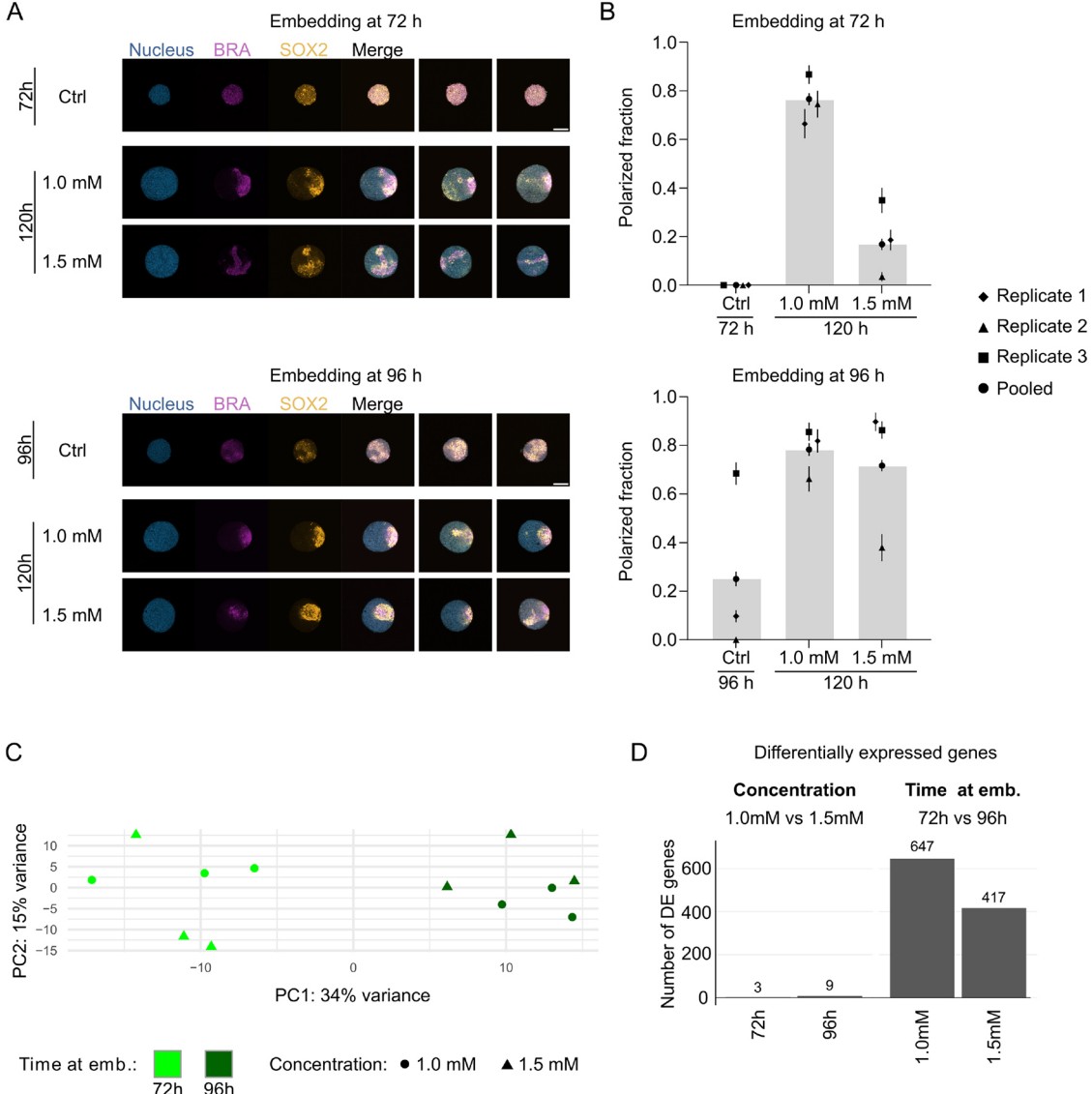

**Fig. 3. Uncoupling of patterning and transcriptional profiles.** (A) Representative immunofluorescence images of gastruloids before embedding (72 h or 96 h) and at 120 h after seeding, following embedding in hydrogels at 72 h (top) or 96 h (bottom). Gel concentrations: 1.0 mM or 1.5 mM. Nuclei (blue), BRA (purple) and SOX2 (orange) are labeled. Scale bars: 100 µm. (B) Quantification of the proportion of gastruloids forming a unique BRA/SOX2 pole at 120 h or at the time of embedding. Data correspond to the conditions shown in A and include embedding at 72 h (top) or 96 h (bottom). Error bars were obtained from bootstrapping (see Materials and Methods). Sample sizes for embedding at 72 h: 72 h Ctrl ($n$=20 for replicate 1; 18 for replicate 2; 22 for replicate 3); 120 h 1.0 mM ($n$=15; 19; 22); 120 h 1.5 mM ($n$=21; 27; 17). Sample sizes for embedding at 96 h: 96 h Ctrl ($n$=22; 23; 22); 120 h 1.0 mM ($n$=21; 21; 22); 120 h 1.5 mM ($n$=19; 21; 23). (C) Principal component analysis (PCA) of bulk RNA-seq experiments, revealing transcriptional differences between gastruloids embedded at 72 or 96 h in hydrogels of 1.0 mM or 1.5 mM concentrations. (D) Number of differentially expressed (DE) genes from bulk RNA-seq, demonstrating minimal transcriptional changes between gel concentrations (1.0 mM versus 1.5 mM) at either embedding time and showing greater differences when comparing embedding times (72 h versus 96 h) within each gel condition. Shown here are representative data and analysis for one experimental replicate. See Fig. S4 for two additional replicates.

singular pole even at higher stiffnesses (Fig. 3A,B, Fig. S7). Notably, ~20% of gastruloids had already established a pole by 96 h (Fig. 3B, Fig. S7D), suggesting that polarization begins either before or very shortly after embedding. These findings indicate that physical constraints applied after polarization is underway do not significantly disrupt this process.

Next, we investigated whether earlier mechanical constraints could impair polarization by embedding gastruloids at 72 h, a time point when no pole is established yet (Fig. 3A, Fig. S7). When embedded in 1.0 mM gels (corresponding to 30 Pa), ~80% of the gastruloids successfully formed a BRA/SOX2 pole by 120 h.

However, this fraction dropped to ~20% in 1.5 mM gels, suggesting that stiffer gels impose sufficient mechanical constraints to impair polarization establishment (Fig. 3B, Fig. S7). Interestingly, gastruloids in 1.5 mM gels were ~10% smaller on average (Fig. S6), potentially indicating increased cellular compression or decreased cell proliferation. However, we could not confirm that cell density or density of dividing cells was higher at 120 h in gastruloids embedded in 1.0 mM versus 1.5 mM gels at 72 h (Fig. S8). These results demonstrate that polarization establishment is sensitive to mechanical constraints during early developmental stages but remains robust once initiated.

To evaluate whether global gene expression was affected by these experimental conditions, we performed bulk RNA-seq. PCA revealed that samples clustered primarily by the time of embedding rather than gel stiffness (Fig. 3C, Fig. S9A,B). The first component (PC1) that separates samples by embedding time explained ~34% of the observed variance in gene expression, whereas gel stiffness had minimal impact. For a fixed embedding time, only a small number of genes were differentially expressed between 1.0 mM and 1.5 mM gels (three genes at 72 h, nine genes at 96 h). In contrast, embedding at 72 h versus 96 h resulted in hundreds of differentially expressed genes, regardless of gel stiffness (647 genes for 1.0 mM, 417 genes for 1.5 mM). These genes included key regulators of embryonic development and stem cell differentiation, such as *Dppa5a*, *Bra* (*T*) and *Hoxa3* (Fig. 3D, Fig. S9C, Table S2).

Surprisingly, the inability of gastruloids embedded at 72 h in 1.5 mM gels to form a BRA/SOX2 pole did not correspond to significant transcriptional changes, as their global gene expression profiles were similar to those of gastruloids that successfully formed poles in 1.0 mM gels. Furthermore, embedding at 72 h induced major transcriptional changes irrespective of whether polarization was maintained (1.0 mM) or disrupted (1.5 mM). This unexpected result contrasts with the assumption that morphological polarization and transcriptional programs are tightly coupled, revealing instead that mechanical constraints can disrupt one without significantly affecting the other. These findings reveal that patterning and transcription can vary independently, providing strong evidence for their uncoupling under specific mechanical conditions.

Together, these results demonstrate that while polarization and transcriptional profiles are generally robust to mechanical constraints, their establishment can be selectively disrupted by the timing and stiffness of embedding. This uncoupling of polarization and gene expression highlights the distinct regulatory mechanisms underlying gastruloid patterning and transcriptional programs.

### Impaired cell motility in dense gel confinement

The minimal differences in transcriptional profiles between gastruloids embedded at 72 h in 1.0 mM and 1.5 mM gels (Fig. 3C,D) contrasted sharply with their differing abilities to form a BRA/SOX2 pole (Fig. 3A,B). Recall that 1.5 mM corresponds to a stiffness of approximately 300 Pa, and that there is an order of magnitude in difference in stiffness between 1.0 mM and 1.5 mM gels (Fig. 1B, Fig. S1A). This discrepancy suggested that defects in BRA/SOX2 pole establishment could arise from differences in cell motility, likely due to the increased mechanical constraints build-up in stiffer gels.

Long-term imaging and tracking of freely floating gastruloids is often hindered by translational and rotational movement during development. Embedding gastruloids in hydrogels, however, significantly reduces movement, even in ultra-soft hydrogels that support elongation (Fig. 4A, Movies 1, 2). This stabilization enables high-resolution imaging without requiring extensive image registration, addressing a major limitation of current imaging workflows. Existing solutions, such as micro-wells or holders, often impose size constraints or require labor-intensive post-processing steps (Beghin et al., 2022; Samal et al., 2020; Oksdath Mansilla et al., 2021; Hashmi et al., 2022). For example, Hashmi et al. (2022) improved live imaging of gastruloids using micro-wells but required the reduction of gastruloid size to ~50 cells, compared to the ~300 cells in the classical protocol optimized for symmetry breaking and axis elongation (van den Brink et al., 2014). By contrast, our hydrogel-embedding approach is compatible with gastruloids of any size or developmental timing, allowing the use of the optimally

defined number of seeded cells to study AP axis formation and produce the full range of cell populations described in the gastruloid system (Bennabi et al., 2024; Fiuza et al., 2024; van den Brink et al., 2014). This user-friendly system facilitates long-term imaging without altering gastruloid development. Using gastruloids with a low proportion of H2B-iRFP-expressing cells, we successfully tracked individual cell movements without the need for complex image registration (Fig. 4B, Fig. S10A,B).

To investigate whether impaired cell migration could explain the defects in BRA/SOX2 pole formation observed in 1.5 mM gels, we conducted live imaging and cell tracking in gastruloids embedded at 72 h in either 1.0 mM or 1.5 mM hydrogels. Imaging was performed using the LS2 Viventis system, with adaptations to the gel embedding protocol to accommodate the sample holder (see Materials and Methods). Gastruloids were generated using a mixture of cells expressing either a *Bra* reporter (TProm-mVenus) or a nuclear marker (H2B-iRFP), enabling cell migration tracking and confirmation of BRA expression patterns.

Tracking over several hours revealed that cells in organoids embedded in 1.0 mM gels appeared more exploratory and migrated further than those in 1.5 mM gels (Fig. 4C), with the latter condition exhibiting significantly lower mean cell migration speeds (Fig. 4D). As the organoids grew during the acquisition period – with more growth observed in 1.0 mM gels – we measured radial growth rates to exclude the possibility that differential growth contributed to the observed differences in cell migration speeds. Radial growth was minimal (1.0 mM: 0.015±0.002 μm/min; 1.5 mM: 0.004±0.006 μm/min) and could not account for the substantial reduction in cell migration speed (Fig. 4D). Surprisingly, other metrics used to characterize cell migration, such as confinement ratio, directional change rate or mean squared displacement (MSD), did not show any significant differences between the two conditions (Fig. S10C-G).

These results suggest that the inability of gastruloids embedded in 1.5 mM gels to establish a BRA/SOX2 pole is not due to transcriptional changes but rather to impaired cell motility and morphogenesis within the mechanically constrained environment. By altering the mechanical properties of the hydrogel, we were able to disrupt polarization and morphogenetic processes while preserving transcriptional profiles, underscoring the crucial role of cell migration in BRA/SOX2 pole formation.

### DISCUSSION

This study addresses a key challenge in developmental biology: disentangling the mechanical and chemical contributions of the environment to gastruloid development, an issue often confounded by the variability and undefined composition of traditional matrices such as Matrigel. By employing a bioinert hydrogel system with tunable stiffness and embedding timing, we provide a precise platform for probing how external mechanical constraints influence developmental processes, including elongation, polarization, and transcriptional regulation.

We demonstrated that gastruloids embedded in ultra-soft hydrogels (<1.0 mM) elongate robustly while maintaining AP axis patterning and transcriptional profiles similar to controls. This finding highlights the capability of our system to support nearly unaffected developmental processes in a mechanically controlled environment. Beyond developmental outcomes, the hydrogel's stability minimizes sample movement during live imaging, enabling precise long-term tracking of gastruloid dynamics. Unlike existing solutions, such as micro-wells, which often impose size constraints or require extensive image registration, our hydrogel platform works

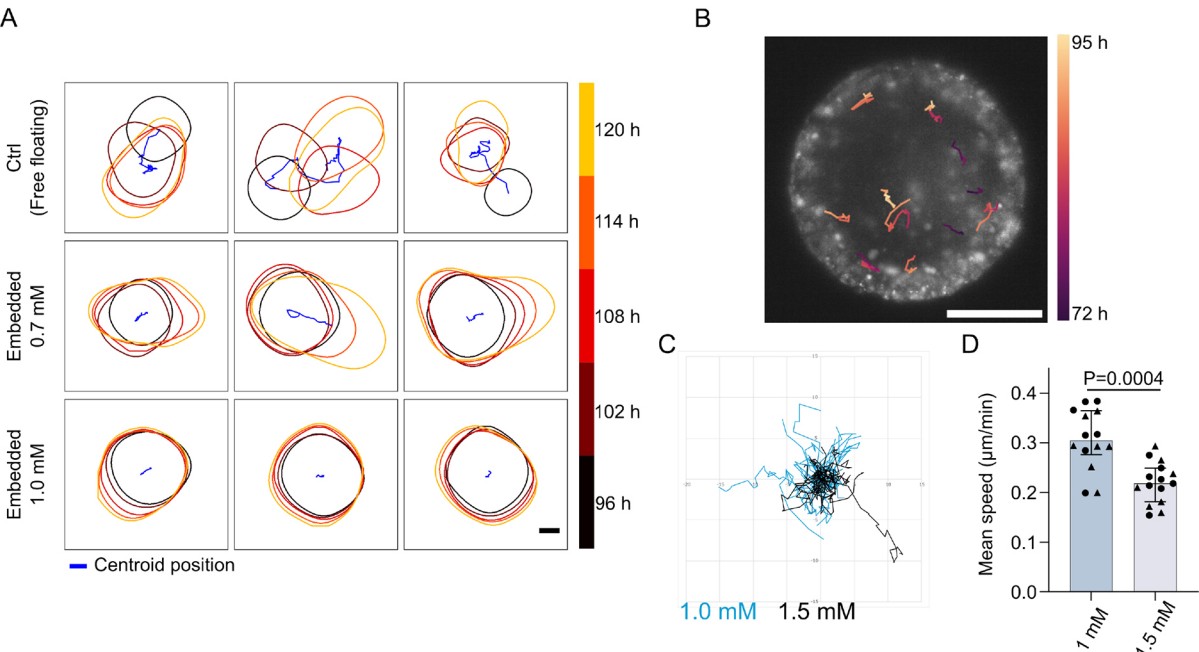

**Fig. 4. Impaired cell motility in dense gel confinement.** (A) Outlines of developing gastruloids from 96 h to 120 h (five time points encoded by color) post seeding under control (Ctrl, freely floating) or embedded (0.7 mM or 1.0 mM) conditions. The trajectory of the gastruloid centroid is shown in blue. Scale bar: 100 μm. (B) Color-coded tracks of cell migration in a gastruloid embedded in a 1.0 mM gel, imaged from 72 h to 95 h post seeding. The image shows H2B-iRFP fluorescence at 95 h post seeding. Scale bar: 100 μm. (C) Trajectories aligned to start at (0,0), from cells expressing H2B-iRFP in gastruloids embedded in 1.0 mM gel (blue) or in 1.5 mM gel. (D) Mean cell speed for tracks with durations ranging from 3 h20 min to 8 h20 min, comparing conditions of 1.0 mM and 1.5 mM gel embedding. Data points represent individual tracks from two gastruloids per condition (triangle and circle symbols). Sample sizes: 1.0 mM, *n*=14; 1.5 mM, *n*=15. Statistics: Mann–Whitney test. Experiment performed in one replicate, with two gastruloids per condition. Error bars represent interquartile range, bar represents median value.

irrespective of gastruloid size or developmental timing, allowing the use of optimal cell numbers for studying axis formation (Van Den Brink et al., 2014; Bennabi et al., 2024; Fiuza et al., 2024). Furthermore, the bioinert nature of our hydrogel enables the disentangling of mechanical and chemical effects, while its modular design allows for functionalization to explore cell–ECM interactions systematically.

A key finding of this study is the decoupling of transcriptional profiles and AP axis patterning under specific mechanical constraints. Embedding in stiffer hydrogels disrupted polarization without altering transcriptional profiles, while earlier embedding significantly affected transcription independently of polarization defects. For example, altering embedding timing impacted the expression of key developmental genes such as *Bra*, without disrupting polarized patterning. This suggests that the transcriptional program proceeds autonomously within each cell, relying on local or short-range cues rather than long-range gradients. Two other studies recently reported alterations in morphogenesis decoupled from transcriptional changes further supporting this view (Mayran et al., 2023 preprint; Bennabi et al., 2024). These findings challenge conventional views of tight coordination between transcription and patterning and raise new questions about how these processes interact under distinct mechanical conditions.

We also identified cell motility as a key factor in polarization, likely mediated through cell sorting and aggregation mechanisms. Gastruloids embedded in stiffer hydrogels (1.5 mM) exhibited significantly reduced cell motility compared to those in softer gels (1.0 mM), potentially explaining their inability to establish a BRA/ SOX2 pole. Increased cell density in stiffer hydrogels may contribute to this reduction in motility. However, additional

factors are likely involved, including altered cytoskeletal dynamics, signaling pathways, and changes in cell adhesion properties (Cermola et al., 2022; Underhill and Toettcher, 2023; de Jong et al., 2024; Anlas et al., 2024; Mayran et al., 2023 preprint). These mechanisms are consistent with the presence of a cell-sorting process, as emphasized in several recent studies (McNamara et al., 2024; Mayran et al., 2023 preprint; Gsell et al., 2025; Oriola et al., 2024 preprint). Together, these findings underscore the importance of cell migration in axis polarization and highlight the value of our platform for dissecting how mechanical constraints shape cell behavior and morphogenesis.

The modularity of our bioinert hydrogel embedding assay enables systematic dissection of the chemical and physical factors influencing gastruloid development. Notably, gastruloids embedded in 1.0 mM gels (30 Pa) – a stiffness close to that of the tissue itself (Oriola et al., 2024 preprint) – did not elongate, whereas robust elongation has been reported in Matrigel at concentrations up to 9 mg/ml (80 Pa) (Van Den Brink et al., 2020). This apparent discrepancy likely reflects differences in matrix properties: our gel is fully bioinert, lacking adhesion sites and resisting degradation, while Matrigel contains ECM components that support adhesion, remodeling, and proteolytic breakdown (Kleinman and Martin, 2005). As a result, the effective stiffness of Matrigel likely decreases over time, whereas our system maintains a stable and well-defined mechanical environment throughout development.

Furthermore, the same hydrogel chemistry allows systematic tuning of stiffness by varying the concentration of reactive groups, enabling broad modulation of mechanical properties while maintaining constant biochemistry. Efficient crosslinking yields dilute yet stable networks, even at low polymer content. A key

physical parameter of the gel is its mesh size, which ranges from approximately 110 nm in the softest gels (0.7 mM) to 24 nm in the stiffest (1.5 mM), based on theoretical estimates (Tsuji et al, 2018). These values remain well above the hydrodynamic radius of most signaling molecules and nutrients (Armstrong et al., 2004), suggesting that passive diffusion is not fully impeded. However, because diffusion coefficients scale with the square of mesh size (Moncure et al., 2022), transport of secreted or exogenous factors is expected to slow down in stiffer gels. These potential diffusion limitations may influence directional cell migration or local signaling dynamics, particularly under the highest stiffness conditions. Nonetheless, our transcriptomic analyses revealed no significant effect of stiffness on global gene expression or cell-type composition, indicating that differences in diffusion likely do not propagate to the level of whole-gastruloid patterning.

While our experiments reveal important insights into the interplay of transcription, patterning and motility, they cannot definitively establish whether gene expression and patterning are entirely independent processes. The observed uncoupling may reflect specific developmental stages or conditions rather than a universal principle. Similarly, the role of motility in polarization warrants further investigation to clarify the causal relationships among these processes. Nevertheless, our hydrogel system provides a robust framework to systematically test these interplays and disentangle the mechanisms underlying early development. By offering precise control over the mechanical environment, this platform opens new avenues for probing the regulatory principles that guide embryogenesis.

## MATERIALS AND METHODS
### Cell culture
mESCs were cultured in 6-well plates (TPP) coated with 0.1% gelatin in water, in a humidified incubator (5% $CO_2$, 37°C). Cell culture media was prepared as follows: DMEM 1X+Glutamax (Fisher, 11584516) supplemented with 10% Decomplemented fetal bovine serum (FBS) (Gibco, 11573397; decomplemented 30 min at 56°C), 1× non-essential amino acids (NEAA; Gibco, 11140-035), 1 mM sodium pyruvate (Gibco, 11360-039), 1% Penicillin-Streptomycin (Gibco, 15140122), 100 μM 2-mercaptoethanol (Gibco, 31350-010), 10 ng/ml leukemia inhibitory factor (LIF; Miltenyi Biotec, 130-099-895), 3 μM CHIR 99021 (GSK3 inhibitor; Sigma-Aldrich, SML1046), 1 μM PD 035901 (MEK inhibitor; Sigma-Aldrich, PZ0162). Cells were passaged every other day (detached using trypsin), and experiments were done using cells that were kept in culture for at least two passages or 5 days after thawing. Cells were tested for *Mycoplasma* contamination using the Eurofins *Mycoplasma* check on a regular basis.

Unless stated otherwise, experiments were performed using the 129/svev mESC line (commercially available from EmbryoMax).

### Cell line generation
The other cell lines used (TProm-mVenus and TProm-mVenus/H2B:iRFP) were generated at EPFL. To generate the Tprom-mVenus cell line, a region of 1392 bp surrounding the brachyury gene was selected to monitor the activity of the brachyury promoter (see Fig. S11). For the transgenic assay, a sequence (see supplementary Materials and Methods) was inserted by Gateway cloning (LR reaction) into the SIF-seq construct (Dickel et al., 2014; Addgene plasmid #51292: pSKB1-GW-hsp68-Venus-H19).

To form the crRNA:tracrRNA duplex, Alt-R CRISPR-Cas9 crRNA (guide sequence: GTTTTAAGATTTCTTTATGG, ordered from IDT) and Alt-R CRISPR-Cas9 tracrRNA (IDT) were hybridized in equimolar concentration in nuclease-free IDTE buffer (44 μM) by incubation at 95°C for 5 min and allowed to cool at 15-25°C on the bench top. The crRNA:tracrRNA duplex were then used to form the RNP complex with a final concentration of 18 μM of Alt-R Cas9 enzyme (previously diluted in Resuspension buffer R from the Neon System kit) and 22 μM of

crRNA:tracrRNA duplex. The RNP complex was incubated for 15 min at 15-25°C.

ESCs (E14tg2a) were thawed and kept for one passage. They were then dissociated and washed twice with PBS. Then, 400,000 cells were resuspended in 22 μl of buffer R (Neon System kit) with 2 μl of RNP complex solution and 4 μg of the SIF-seq construct containing the 1392 bp near the brachyury promoter (indicated above). Cells were electroporated with the Neon electroporation system (with 1100 V, 20 ms and 2 pulses settings) and seeded in a 6 cm Petri dish with 3 ml of DMEM media. Media was renewed the next day. Medium was replaced 24 h later by DMEM complemented by adding 1× HAT selection supplement (Gibco) media and was renewed daily for 5 days. Seventeen clones were picked and allowed to recover for 1 week in DMEM medium supplemented with HT. These were analyzed by PCR (mVenus FWD: caccatggtgagcaagggcgag; mVenus REV: ttctgctggtagtggtcggcga; Ampicillin FW: ctgcaactttatccgcctcc; Ampicillin REV: gtgcacgagtgggttacatc).

Clones positive for the presence of mVenus and the absence of ampicillin were amplified and independently verified.

To add H2B:iRFP fluorescent protein, 2 μg of pCAG-H2BtdiRFP-IP (gift from Maria-Elena Torres-Padilla, Addgene plasmid #47884; RRID: Addgene 47884) was transfected in 300,000 brachyury promoter reporter E14tg2a using FuGENE (Promega). Media was changed the following day, and 24 h later puromycin selection was performed over 5 days. The pool of resistant colonies (>1000 colonies) was allowed to grow and was passaged as a pool.

### Gastruloid generation
Gastruloids were generated as described by Beccari et al. (2018). The N2B27 medium was prepared every 3 weeks in-house using 250 ml DMEM/F12+GlutaMax (Gibco, 10565018), 250 ml Neurobasal (Gibco, 21103049), 2.5 ml N2 (Gibco, 17502-048), 5 ml B27 (Gibco, 17504-044), 1× NEAA (Gibco 11140035), 1 mM sodium pyruvate (Gibco, 11360-039), 100 μM 2-mercaptoethanol (Gibco, 31350-010), 1% Penicillin-Streptomycin (Gibco, 15140-122), 2.5 ml Glutamax (Gibco, 35050061). Gastruloids were generated by manually seeding 300 cells per well in Costar Low Binding 96-well plates (Corning, 7007), in a volume of 40 μl per well. After 48 h of aggregation, spheroids were exposed to a 24 h pulse of Wnt agonist by adding 150 μl of 3 μM CHIR 99021 (CHI in the text) in N2B27 to each well, unless stated otherwise. N2B27 media was then changed every 24 h by replacing 150 μl of media per well until 120 h. For gastruloids in hydrogels, the media was also replaced every 24 h.

### Gel embedding
Gastruloids were embedded in hydrogels 72 h or 96 h after seeding, then left to grow until 120 h. For gastruloid embedding, 20-30 gastruloids were collected from a 96-well plate and left to sediment in a Falcon tube. Meanwhile, gel components were prepared, following the proportions of the manufacturer of the 3-D Life Dextran-PEG Hydrogel FG (Cellendes, FG90-1, containing the components PEGLink, Dextran-Maleimide and CB Buffer pH 5.5) to reach the indicated function concentrations, setting 50% of gastruloid suspension and a total volume of 120 μl per gel. For each gel, two tubes were prepared: one tube containing ultrapure water and PEG-Link, and one tube containing CB Buffer (pH 5.5) and gastruloid suspension in N2B27. Dextran-Maleimide was added to the second tube at the last moment, followed by a quick homogenization with the pipette, pooling of the two solutions, homogenization, then deposition of the gel mix into a glass-bottom dish (Cellvis, D35-10-1.5-N). Quickly, a membrane was deposited on top of the gel (Isopore filter, 5.0 μm membrane, 13 mm diameter; Merck, TMTP01300), followed by an adhesive ring (Delta microscopies, slide wells D70366-12) to secure the membrane. The dish was then put in the incubator for 15 min for gel formation, after which 2 ml of N2B27 was added on top, and the embedded gastruloids were kept for culture as usual. For Matrigel embedding, gastruloids were mixed with Matrigel GFR (Corning, 356230, Lot 2187005, protein concentration 13.5 mg/ml) to obtain 120 μl at the final protein concentration of 1.0 mg/ml or 2.5 mg/ml. The mixture was then poured similarly to the dextran gel mix in a dish and covered with a secured membrane and placed in the incubator. N2B27 was added after 15 min. The same Matrigel was used for the Matrigel-supplemented N2B27.

## Gel characterization

Hydrogel stiffness was measured using a rheometer (Kinexus Ultra, Malvern). Briefly, gel formation was measured by preparing the gel mix as described above, only replacing the gastruloid suspension by N2B27 media. Upon mixing, 100 µl of gel mix was deposited between the rheometer geometry (flat 20 mm diameter tool) and plate, kept at 4°C. Excess of liquid was removed. The temperature was quickly ramped up to 37°C while oscillating rotations of the mobile geometry measured the gel stiffness. Frequency and amplitude were fixed at 3 Hz and 5%. After 15 min, N2B27 was added in contact of the gel to simulate the effect of adding medium in the usual experiment. Gelling dynamics could be observed through the shear modulus measurement in live, as shown in Fig. S1. A stable plateau value was reached within 30 min.

## Bulk RNA-seq sample preparation and analysis

### Sample generation

Gastruloids were generated and embedded as described above, and as a negative control, samples where the mESC aggregates were not submitted to a CHI pulse and left in a 96-well plate were generated. For each replicate, about 30 gastruloids were processed per condition. At 120 h after seeding, dishes containing embedded gastruloids were treated as follows: the adhesive ring was lifted, and a solution of 1:20 Dextranase (Cellendes, D10-1) in PBS ($Ca^{++}$/$Mg^{++}$) was added to each dish, and left >20 min in the incubator until gastruloids were freely moving and could be collected in a falcon tube. Gastruloids in 96-well plates (control and negative control) were collected into Falcon tubes. All gastruloids were then washed twice with cold PBS ($Ca^{++}$/$Mg^{++}$) before being snap-frozen in liquid nitrogen, and stored at −80°C until RNA extraction.

### Sample processing

The RNeasy Mini kit (QIAGEN) with on-column DNase digestion was used for RNA extraction following the manufacturer's instructions. RNA quality was assessed on a TapeStation TS4200, all RNA samples showed a quality number (RIN) above 9. RNA-seq library preparation with Poly-A selection was performed with 550 ng of RNA using the Illumina stranded mRNA ligation and following the manufacturer's protocol (1000000124518 v.01). Libraries were quantified by qubit DNA HS and profile analysis was carried out on a TapeStation TS4200. Libraries were sequenced on a Novaseq 6000, with paired-end 75 bp reads.

### Bulk RNA-seq analysis

RNA-seq preprocessing was performed using a local installation of Galaxy (The Galaxy Community, 2024). Adapter and bad quality bases were removed from fastq files using cutadapt version 4.4 (Martin, 2011) (-q 30 -m 15 -a CTGTCTCTTATACACATCTCCGAGCCCACGAGAC -A CTGTCTCTTA-TACACATCTGACGCTGCCGACGA). Filtered reads were aligned on mm10 using STAR version 2.7.10b (Dobin et al., 2013) with the ENCODE parameters and a custom gtf (customized gtf file from Ensembl version 102 mm10.: doi:10.5281/zenodo.7510406). FPKM were computed with cufflinks version 2.2.1.3 (Trapnell et al., 2010; Roberts et al., 2011) using –max-bundle-length 10,000,000 –multi-read-correct –library-type "fr-firststrand" -b mm10.fa –no-effective-length-correction -M mm10 chrM.gtf. For analyses, genes from mitochondrial genes were excluded. For both analyses (concentration effect and time effect), the fragments per kilobase per million mapped fragments (FPKM) values were transformed with log2(1+FPKM), and the 500 genes with the highest variance were selected. PCA was computed on these genes, and clustering was performed using 1−Pearson's correlation coefficient as distances with Ward.D2 method. Pairwise differential expression analysis was computed with DESeq2 (Love et al, 2014) on raw counts from STAR (excluding mitochondrial genes). A gene was considered as DE when the adjusted $P$-value was below 0.05 and the absolute log2 fold-change was above 1. For deconvolution, the post-process bioRxiv version RDS (R Data Serialization) file from the wild-type samples from Mayran et al. (2023) was downloaded from Gene Expression Omnibus (GEO) (GSE247509). Only cells with time at 120 h were kept. Only fates representing at least 0.2% of cells were considered. To get count values as close as possible to the 3′ counting from single-cell RNA-seq, the STAR counts from the bulk RNA-seq were converted to FPKM using EdgeR version 4.4.2. The deconvolution was computed with DWLS version 0.1.0 Tsoucas et al. (2019).

## Transmitted light imaging for morphological characterization

At 120 h after gastruloid seeding, embedded gastruloids or gastruloids cultured in 96-well plates were imaged in transmitted light using an Olympus video microscope with Olympus CellSens dimension 3.1 software, equipped with a Hamamatsu C11440-36U CCD camera with a pixel size of 5.86×5.86 µm and a 4×0.13 NA objective. Gastruloids in 96-well plates were imaged individually, whereas gastruloids in gels were imaged by tiling over the whole region of the glass-bottom dish, then performing stitching with Fiji (Schindelin et al., 2012). Embedded gastruloids could then be cropped from the large region to obtain individual TIFF images of embedded gastruloids.

Images were then processed in Python to obtain a mask of each gastruloid and measure morphological characteristics such as the gastruloid length [obtained by computing the medial axis and extending it to the organoid extremities, as described by Merle et al. (2023)], the straightness ratio (ratio between the gastruloid length and the distance between the body axis extremities), and the aspect ratio. Elongation index was computed based on the same masks obtained with the Python workflow, using a version of the ImageJ macro published by Girgin et al. (2021), adapted to work directly on masks.

## Gastruloid immunostaining

For this protocol, only PBS ($Ca^{++}$/$Mg^{++}$) was used, and will be called PBS in this section. Any tube, plate or pipette tip that contained gastruloids was either low binding or coated with PBSF (PBS, 10% FBS). Washes were carried out by spinning the gastruloids for 1 min at 10 $g$ to help sedimentation, and aspirating the liquid.

Gastruloids cultured in a 96-well plate until 120 h were collected in a low-binding 15 ml Falcon tube and washed with PBS, before proceeding with 2 h fixation in 4% paraformaldehyde in PBS at 4°C. Meanwhile, embedded gastruloids were fixed as follows: N2B27 was removed from the dishes, and gels were washed with PBS before fixation in 4% paraformaldehyde in PBS at room temperature (RT) for 2 h 30 min. After fixation, samples were washed twice for 15 min each wash in PBSF at RT, and once in PBS.

Embedded gastruloids were de-embedded after this step: the adhesive ring and membrane were lifted, and dishes were incubated with 1:20 Dextranase (Cellendes, D10-1) in PBS at 37°C for >20 min, until gastruloids could freely move in the dish. Gastruloids were then recovered in a low-binding Falcon tube and washed with PBS.

All gastruloids were then permeabilized by incubating twice for 30 min in 13 ml of PBSFT (PBSF with 0.03% Triton X-100) at RT. Primary antibody staining was performed overnight at 4°C by incubating in a solution of primary antibodies with 1:500 DAPI (Sigma-Aldrich, D9542-5MG), in 500 µl PBSFT (per condition). The next day, gastruloids were washed twice for 20 min at RT in PBSF, once in PBS, then in PBSFT. They were then incubated overnight at 4°C with secondary antibodies, in 500 µl of mix containing 1:500 DAPI in PBSFT. Primary antibodies used were: rat anti-SOX2 (1:200; eBioscience, 14-9811-80), rabbit anti-brachyury (1:200; Abcam, ab209665), rabbit anti-FOXC1 (1:500; Abcam, ab223850). Secondary antibodies used were: donkey anti-rabbit Alexa Fluor 647 (1:500; Invitrogen, A-31573), donkey anti-rat Alexa Fluor 488 (1:500; Invitrogen, A-21208), goat anti-rat Alexa Fluor 647 (1:500; Invitrogen, A21247), goat anti-rabbit Alexa Fluor 488 (1:500; Invitrogen, A11070).

Finally, gastruloids were washed twice for 20 min each wash at RT in PBSF, then in PBS. They were then transferred to a 6-well plate filled with PBS using a cut P1000 tip, to finish washing the gastruloids and remove debris or impurities. Subsequently, gastruloids were transferred to 1.5 ml Eppendorf tubes, and all PBS was removed before resuspending in 150 µl of mounting media (50/50 PBS/Aquapolymount; Polysciences 18606-20), and transferring them to a glass-bottom dish (Cellvis, D35-10-1.5N). Gastruloids could then be moved to scatter them across the dish, and were left to sediment at the bottom of the dish for >24 h at 4°C, after sealing dishes with Parafilm to minimize evaporation. Samples were then sealed with a cover glass and nail polish for conservation.

For the staining of H3S10P, the protocol was modified to improve in-depth staining and imaging. Permeabilization was performed in PBS+10% FBS+0.3% Triton X-100 at RT. Primary antibody staining was performed

by incubating overnight at RT in a solution of 1:500 DAPI (Sigma-Aldrich, D9542-5 mg) and 1:1000 rabbit anti-phosphohistone H3 (pSer10) (Sigma-Aldrich, H0412) in PBS+10% FBS+0.3% Triton X-100. The next day, gastruloids were washed twice for 20 min each wash at RT in PBSF, once in PBS, then in PBS+10% FBS+0.3% Triton X-100. They were then incubated overnight at RT in a solution of 1:500 DAPI (Sigma-Aldrich, D9542-5 mg), 1:500 donkey anti-rabbit Alexa Fluor 647 (Invitrogen, A-31573) and 1:500 donkey anti-rat Alexa Fluor 488 (Invitrogen, A-21208). Finally, gastruloids were washed twice and processed as for the previous mounting protocol, except for the mounting media that was replaced by RapiClear 1.47 (SunJin Labs, RC147001).

### Confocal imaging of immunofluorescence samples

Immunofluorescence samples for combined BRA and SOX2 staining were imaged using a Zeiss LSM980 inverted laser scanning confocal microscope controlled with Zen 3.3 software (Zeiss), and equipped with a 10×0.45 NA air objective (Zeiss). Images were acquired as $z$-stacks, by taking 30 slices in a 150 μm range, resulting in a voxel size of 0.22×0.22×5.00 μm. Samples were illuminated using 405/488/639 lasers diodes successively. Signal for BRA (Alex Fluor 647, detection 641-693) and SOX2 (Alexa Fluor 488, detection 509-632) was collected on two separate GaAsP-PMT detectors, and for DAPI (detection 408-501) on a Multialkali-PMT detector.

Immunofluorescence samples for combined FOXC1 and SOX2 staining were imaged using a Nikon AX inverted laser scanning confocal microscope controlled with NIS-Elements AR 5.42.06 software (Nikon), and equipped with a 10×0.45 NA air objective (Nikon) and 405/488/561/640 lasers. Signal for FOXC1 (Alexa Fluor 488, detection 503-541), SOX2 (Alexa Fluor 647, detection 653-726) and DAPI (detection 420-516) were collected on GaAsP detectors. Images were acquired as $z$-stacks, by taking 30 slices in a 150 μm range, resulting in a voxel size of 0.216×0.216×5.00 μm. Confocal imaging of samples stained for phospho-histone H3 (pSer10) was performed on the same device, using a 20×0.80 NA air objective (Nikon) and 405/488/640 nm lasers. Signals for H3S10P (Alexa Fluor 488, detection 503-541) and DAPI (detection 420-516) were collected on GaAsP detectors. Images were acquired as $z$-stacks, taking 17 slices with a voxel size of 0.192×0.192×10 μm.

### Density analysis

Density of cells and density of H3S10P$^{high}$ cells were determined from single-plane images of gastruloids stained with DAPI and anti-phospho-histone H3 (pSer10). Image analysis was carried out in ImageJ. Briefly, in a single plane approximately 80 μm deep in the gastruloid, average DAPI intensity was measured within the gastruloid mask to evaluate cell density in the organoid, and H3S10P$^{high}$ cells were detected and counted by filtering the image with a Gaussian (sigma=5), setting a threshold using the MaxEntropy method, performing a watershed then counting objects. Together with the area of the gastruloid slice, this gave access to the density of dividing cells.

### Extraction of 1D gene expression profile

Intensity profiles along the antero-posterior axis were computed as described by Merle et al. (2023). All analyses were performed on maximum intensity projections of confocal image stacks. Using a custom Python script, masks of gastruloids were generated from the DAPI channel. The main body axis of the gastruloid was then defined by finding the medial axis and expanding its extremities with straight lines, tangent to the medial axis ends, until the lines cross the gastruloid contour. The contour was cut at these intersection points, and each side was subdivided in 100 equidistant points. Sections of the gastruloids could be defined by connecting pairs of equivalent points on each side. To obtain the intensity profiles of fluorescent signals, the average intensity in each bin was measured. Since the fixation protocol is different for embedded and non-embedded gastruloids, profiles were compared by shape, and not fluorescence intensity. Specifically, intensity values were scaled for each experiment relative to the average profile of all gastruloids, using the 10% lowest and highest values for reference. Spatial profiles were normalized to the medial axis length of individual gastruloids. Boundary positions for gene expression profiles were defined as the relative position along the medial axis of the gastruloid where

the half-maximal expression level within the boundary regions is reached, as described by Bennabi et al. (2024).

### Live imaging of gastruloid elongation and pattern formation by video microscopy

Live imaging of brachyury pattern dynamics and gastruloid elongation with the TProm-mVenus cell line were performed with a wide-field inverted fluorescence microscope (IX81, Evident) using a 20× objective (UPLFLN20X). Gastruloids were embedded in 0.7 μM dextran-gels at 96 h. For each embedded gastruloid, $z$-stacks with 20 μm spacing were acquired every 30 min in brightfield and fluorescence. Gastruloid contour and medial axis were extracted from brightfield images and intensity profiles were extracted along this axis.

Movies of gastruloid chimeras composed of 129/svev cells and cells expressing a nuclear marker H2B-iRFP were acquired using a wide-field inverted fluorescence microscope (IX81, Evident) using a 20× objective (UPLFLN20X). Gastruloids were embedded in 0.8 μM dextran-gels at 96 h. For each embedded gastruloid, $z$-stacks with 20 μm spacing were acquired every 30 min in brightfield and fluorescence. Two gastruloids from the same experiment are shown.

### Cell tracking

Movies for single-cell tracking were acquired on gastruloid chimeras composed of TProm-mVenus and TProm-mVenus/H2B:iRFP. Images were acquired every 20 min starting from 72 h, using a LS2 Viventis light sheet microscope with its 25× objective configuration. This resulted in a voxel size of 0.26×0.26×3 μm. Cell tracking was performed by manual tracking using TrackMate in Fiji software in 3D (Schindelin et al., 2012). Gels were cast in the LS2 Viventis sample holders (SHT SW0.8) without a membrane deposited on top. Movies from two gastruloids embedded in a 1.0 mM gel and two gastruloids embedded in a 1.5 mM gel were analyzed, with only trajectories of 10-25 time points being considered.

Tracks were then analyzed by both extracting features from TrackMate (mean speed, confinement ratio, mean directional change rate) and computing the MSD and its associated metrics (diffusion coefficient, diffusion exponent) in Python.

### Statistical analysis

Graphs and statistical analysis were generated using GraphPad Prism version 10.5.0 for Windows (GraphPad Software; https://www.graphpad.com) or Python. All replicates are biological replicates. In bar graphs, median±IQR are plotted. For profiles, mean±s.e.m. are plotted. For box and whiskers plots, the line represents the median, the box the first and third quartiles, the whiskers the minimum and maximum values.

### Genome browser view of the brachyury promoter

Bigwig files with RNA-seq profiles of control gastruloids of 48 h to 120 h were retrieved directly from GEO (GSE247508). Bigwig files with ATAC-seq profiles of wild-type gastruloids of 48 h to 120 h were retrieved directly from GEO (GSE247507).

The plot (Fig. S11) was generated with pyGenomeTracks (Lopez-Delisle et al., 2020) version 3.9 on mm10:chr17:8,428,652-8,442,571.

### Acknowledgements

We thank I. Bennabi, M. Cerminara, C. Chureau, L. Friedman, P. Hansen, M. Lutolf, V. Marthiens and M. Merle. We thank the Gene Expression Core Facility at EPFL for generation of RNA-seq libraries and sequencing, and the HPC core facility at Institut Pasteur. We acknowledge the use of AI-based writing tools for language enhancement. The authors subsequently reviewed and edited the content as necessary and take full responsibility for the publication's final content.

### Competing interests

The authors declare no competing or financial interests.

### Author contributions

Conceptualization: J.W.-N., A.S., S.G., T.G.; Data curation: J.P., J.W.-N., A.M., L.L.-D., P.O.; Formal analysis: J.P., J.W.-N., A.M., L.L.-D.; Funding acquisition: D.D., T.G.; Investigation: J.P., J.W.-N., A.M., A.S., T.G.; Methodology: J.P., J.W.-N., A.M., L.L.-D., P.O., S.G.; Project administration: T.G.; Resources: D.D., S.G., T.G.; Software: J.P., J.W.-N., A.M., L.L.-D., P.O.; Supervision: S.G., T.G.; Visualization: J.P., J.W.-N., A.M., L.L.-D., P.O., T.G.; Writing – original draft: J.P., J.W.-N., T.G.; Writing – review & editing: J.P., J.W.-N., A.M., L.L.-D., T.G.

## Funding

This work was supported by Institut Pasteur (particularly the HPC core facility), Centre National de la Recherche Scientifique, Agence Nationale de la Recherche (ANR-20-CE12-0028'ChroDynE', ANR-23CE13-0021'GastruCyp' and ANR-10 LABX-73'Revive'), and by funding from the European Research Council (ERC-2023-SyG, Dynatrans, 101118866). In addition, the École Polytechnique Fédérale de Lausanne also supported this work, the Swiss National Science Foundation (SNSF; Schweizerischer Nationalfonds zur Förderung der Wissenschaftlichen Forschung; grants 310030-196868, CRSII5-189956 and 407940-206405), and the Human Frontier Science Program (HFSP LT000032/2019-L). Open Access funding provided by Princeton University. Deposited in PMC for immediate release.

## Data and resource availability

The bulk RNA-seq data have been deposited to GEO under accession number GSE288158. The code used to produce the next-generation sequencing analysis (bulk RNA-seq) can be found on: https://github.com/lldelisle/allRNAseqScriptsFrom PineauWongNgEtAl2025. All other relevant data and details of resources can be found within the article and its supplementary information.

## The people behind the papers

This article has an associated 'The people behind the papers' interview with some of the authors.

## Peer review history

The peer review history is available online at https://journals.biologists.com/dev/lookup/doi/10.1242/dev.204711.reviewer-comments.pdf

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
