## [Peer Review File · Development (Cambridge, England)]

Fine-tuning mechanical constraints reveals uncoupled patterning and gene expression in murine gastruloids

Judith Pineau, Jerome Wong-Ng, Alexandre Mayran, Lucille Lopez-Delisle, Pierre Osteil, Armin Shoushtarizadeh, Denis Duboule, Samy Gobaa and Thomas Gregor
DOI: 10.1242/dev.204711

Editor: Matthias Lutolf

Review timeline

Original submission: 7 February 2025

Editorial decision: 8 April 2025

First revision received: 17 July 2025

Accepted: 20 August 2025

Original submission

First decision letter

MS ID#: dev.204711

MS TITLE: Fine-tuning mechanical constraints uncouples patterning and gene expression in murine pseudo-embryos

AUTHORS: Judith Pineau, Jerome Wong-Ng, Alexandre Mayran, Lucille Lopez-Delisle, Pierre Osteil, Armin Shoushtarizadeh, Denis Duboule, Samy Gobaa and Thomas Gregor

Dear Dr Gregor,

I have now received all the referees' reports on the above manuscript, and have reached a decision. The referees' comments are appended below, or you can access them online: please go to:

As you will see, the referees express considerable interest in your work, but have some significant criticisms and recommend a substantial revision of your manuscript before we can consider publication. If you are able to revise the manuscript along the lines suggested, which may involve further experiments, I will be happy receive a revised version of the manuscript. Your revised paper will be re-reviewed by one or more of the original referees, and acceptance of your manuscript will depend on your addressing satisfactorily the reviewers' major concerns. Please also note that Development will normally permit only one round of major revision. If it would be helpful, you are welcome to contact us to discuss your revision in greater detail. Please send us a point-by-point response indicating your plans for addressing the referees' comments, and we will look over this and provide further guidance.

Please attend to all of the reviewers' comments and ensure that you clearly highlight all changes made in the revised manuscript. Please avoid using 'Tracked changes' in Word files as these are lost in PDF conversion. I should be grateful if you would also provide a point-by-point response detailing how you have dealt with the points raised by the reviewers in the 'Response to Reviewers' box. If you do not agree with any of their criticisms or suggestions please explain clearly why this is so.

Reviewer 1

Advance summary and potential significance to field

In this manuscript, Pineau and Wong-Ng et. al., present new insights on mechanobiology of mouse gastruloid development. The authors utilize biologically inert dextran hydrogels to study the effect of extracellular stiffness on gastruloid morphogenesis. They report that mouse gastruloids could elongate and establish anteroposterior patterning when embedded in ultra-soft hydrogels and failed elongation in stiffer gels might be due to reduced cell motility. They demonstrate that matrix stiffness does not have a significant influence on the transcriptome of gastruloids and highlight that the timing of gel embedding is critical in inducing changes at the bulk RNA level. Overall, the authors describe this work as evidence of developmental decoupling of morphogenesis and transcription.

Comments for the author

The authors state that "The minimal transcriptional and morphological deviations observed suggest that gastruloids grown in these conditions can be used interchangeably with those grown in standard culture". Although morphology and transcriptome at bulk level seems largely unchanged, tissue organization of control and embedded gastruloids could be different. The authors should support this claim by performing single cell RNA sequencing to better understand the cell type compositions of elongating and non-elongating gastruloids. Moreover, immunostainings or in situ hybridizations to show spatial arrangement of key tissues present in gastruloids (presomitic mesoderm, endoderm, cardiac mesoderm) are necessary.

As cited in the manuscript, mouse gastruloids embedded in Matrigel elongate efficiently. The authors should include this control condition. It would be interesting to see whether Matrigel embedding promotes straighter gastruloids as in dextran hydrogels.

The authors compare mechanical properties of dextran hydrogels to Matrigel and show that 1mM dextran hydrogels have lower shear and elastic modulus than 9 mg/ml Matrigel. However, gastruloids embedded in 1mM dextran hydrogels do not elongate. How would the authors explain this? Can it be due to the rich extracellular matrix protein composition of Matrigel? Is it possible to introduce ECM proteins (laminin, collagen, nidogen) into dextran gels by simply mixing with Matrigel? Would this promote formation of somitic structures? The authors should perform these experiments to better understand the dynamics of axial elongation.

The authors offer gel embedding as a better set up for live imaging, however they do not provide timelapse videos of elongating gastruloids. Providing such videos would strengthen this claim.

What is the mesh size of the ultra-soft and stiff gels? Can it be possible that in addition to physical constraints, stiffer gels also entrap secreted molecules that might have influence on axial elongation?

A major drawback of conventional mouse gastruloid culture is the bending and loss of elongated morphology at the later stages of culture. It is particularly interesting to see dextran embedded gastruloids adopt a straighter morphology than controls. If the hydrogels are stable, would these "straighter" gastruloids maintain the elongated morphology up to 168h? If not, can they be freed of the hydrogel at 120h and maintained in suspension longer?

Early embedding seems to prevent elongation. Would embedding at later timepoints (120h) increase longevity of gastruloids?

Increased cellular compression and decreased cell proliferation could be quantified by YAP and Ki67(or EdU) stainings in gastruloids embedded at 72h in 1.5 mM gels.

In discussion, the text reads "Gastruloids embedded in stiffer hydrogels (1.5mM) exhibited significantly reduced cell motility compared to those in softer hydrogels (1.0 mM)" however the related Fig. 4C does not report any significance. Authors should either replicate the experiment to test significance or modify the text. Overall, it would be more relevant to plot all replicates of morphological quantifications (length, straightness ratio) in the same graphs and report statistical

significance. This is missing throughout the paper. Supp. Fig. 5D attempts to do this but misses statistical analysis.

Supp. Fig. 7a: What do the colors indicate? Is it the depth of the location the cell is found? It is difficult to judge without a legend.

Supp. Fig. 7b: Do different colors indicate individual cells? Which cells are BRA+ and which are H2B+? It would help a lot if a mask of gastruloid is also displayed, similar to S7a.

Reviewer 2

Advance summary and potential significance to field

In this work Pineau et al. explore the effect of hydrogel embedding on the formation of mouse 3D gastruloids. To do this, the authors use dextran-based PEG hydrogels with varying stiffnesses. When gastruloids are embedded in ultra-soft hydrogels (<30 Pa), they are able to polarize and elongate normally as in control non-embedded gastruloids. However, when the stiffness is increased (>30 Pa), embedding at 96h impairs elongation while leaving polarization unaffected. In contrast, if hydrogel embedding is done at 72h, for very stiff gels (~300 Pa), both polarization and elongation are impaired. Finally, in this last condition, the authors track cell movement and report impaired cell motility.

The work is sound and the manuscript is well written. In particular, I believe the PEG hydrogel embedding method is of special interest to the gastruloid community. Coincidentally, the fact that embedding in ultra-soft hydrogels improves cell tracking without affecting gastruloid development is also an interesting result of the paper. Below I enumerate a list of major and minor comments which I think can significantly improve the manuscript. Provided that my comments are successfully addressed, I am happy to recommend the manuscript for publication in Development.

Major comments

1) The title of the manuscript states that "fine-tuning mechanical constraints uncouples patterning and gene expression". I do not understand how the authors reach this conclusion from the obtained results. In particular, in their discussion they say that "[...] the transcriptional program proceeds autonomously within each cell, relying on local or short-range cues rather than long-range gradients. These findings challenge conventional views of tight coordination between transcription and patterning". I believe the message the authors are trying to convey is that their results challenge a reaction-diffusion mechanism (long-range cues) and support a cell sorting mechanism (short-range cues) for symmetry breaking. However, I find the title misleading, suggesting that mechanical embedding actively uncouples patterning and gene expression, and thus when the system is not embedded patterning and gene expression are coupled.

2) The hydrogel stiffness increases logarithmically in the mM range of linking functions as shown in Fig. S1C,D. As reported in the paper, the major effect on embedding is found around ~30-300 Pa (1 and 1.5 mM), however this is an order of magnitude in range. The uncertainty in stiffness for the 1 mM conditions seems to be 30 ± 10 Pa from Fig. S1C. Given that the dispersion in gel stiffnesses is of ~10 Pa, my question is why the authors did not explore in more detail the range of 10-100 Pa. Indeed, recent work has reported that the stiffness of gastruloids is ~30 Pa (see Oriola et al. bioRxiv 2024) and thus this would be the interesting range to study rather than very large stiffnesses (300 Pa).

3) In Fig 4C, the authors quantify what they call "cell migration speed" and find a ~30% decrease in motility from 1 mM to 1.5 mM hydrogels. Taking into account this is an order of magnitude change in stiffness (from 30 to 300 Pa), I find the change in cell motility not very dramatic. Indeed, one could expect that the mean cell motility is not greatly affected but the diffusion coefficient is due to caging effects. Therefore, I think it would be instructive to also compute the relative mean squared displacement to explore the type of anomalous diffusion found in both conditions, which might be more meaningful than the mean cell speed.

Minor comments

1) In order to improve the readability of the paper I would suggest to include Fig. S1 as Figure 1 on the main text. I think this is necessary for the reader to clarify details on the method and the conditions used. In addition, panels Fig. S1C and D I think are really important and should be in the main text.

2) In Fig. 1 it is shown that ultra-soft hydrogels (1-5 Pa) are sufficient to straighten gastruloids however only when the stiffness is around ~30 Pa elongation is impaired. Isn't this result somehow implying that gastruloids are easier to bend than to stretch? This would be an interesting result to discuss.

3) The authors suggest that the inability of gastruloids to establish a BRA/SOX2 pole when embedded at 72h in 1.5 mM hydrogels is due to impaired cell motility and not due to transcriptional changes. They reason that this is likely due to an increase in cell density which is also supported by the fact that the aggregates are smaller in this condition. This would point towards a cell sorting mechanism for pole formation which is in line with recent work (McNamara et al. NCB 2024, Gsell et al. 2025, Oriola et al. bioRxiv 2024). It would be interesting to add this in the discussion.

4) From the methods section I understand that cells are cultured in 2i LIF conditions. I would include this explicitly in Fig. S1 in panel A.

Reviewer 2

Advance summary and potential significance to field

Pineau et al are asking a fundamental question: how does the balance between mechanical and genetic programmes influence morphogenesis and cell fate? To answer this, they couple the Gastruloid model system with hydrogels that have been tuned to specific stiffness. Critically this approach is a good diversion from the typical Matrigel approaches which, although providing different stiffnesses, can't be tuned and aren't inert.

They find that whereas low stiffness hydrogels can produce the typical elongations and gene expression patterns, stiffer hydrogels maintain expression but not polarisation. In addition, the timing of hydrogel embedding is also important, altering the transcriptional profiling. Live imaging approaches add a lovely dimension to this work, and allows a mechanistic understanding of these differences, suggesting that cell mobility is a critical input into these processes.

As a quick summary, I think this manuscript is exactly what the gastruloid system should be used for: uncoupling the different inputs that cells and embryos would receive during development in a proper quantitative and methodological manner, understanding what input regulates what process. The data are clear, and the conclusions match what is presented, although it would be good to see statistics used in the quantification of gastruloid metrics (see below). The work is a very important addition to the field, really show-cases how useful the system is, and was a pleasure to read. I only have minor comments.

Comments for the author

Specific comments

1. For Fig. 1, I like the measure of gastruloid straightness; as far as I can remember, this hasn't been done before, even though many of us in the field mention the different morphologies in passing. Do the authors think elongation index could also be used here? I think it might add an additional dimension which is straightforward to do.

2. In addition to Fig. 1, it might be good not only to say the N, but also the number of replicate experiments in the figure for each time-point (only in the legend, not the figure); it just adds more weight to the power of gastruloids in getting replicate experiments compared with animals. I know you show in the supplemental the three replicates, but a quick note of this in the legend is fine.

3. On page 4, the gene names are in capitals... I think this ought to be changes to lower-case with the exception of the first letter.
4. The sentence "fluorescence intensities were normalized for both intensity and length, enabling a comparison of the spatial expression profiles..." is interesting, important, and probably should be an echo what's in the M&M section just so we don't have to hunt through the manuscript to find what this is normalisation is
5. Figure 2: This is a lovely figure, but two things I think need modifying slightly. It might be useful if (either here or supplemental) the authors show maybe one gastruloid with the colours split so you can see where they are? I really like seeing the three replicate stains, but it could be useful to see the different channels somewhere. In B and C, it would be good if the legend spanned the two graphs so it's easier to see that it belongs to both; also I had to check and see whether you included the no chi quantification. Should this be in here, as it's coloured 'red' and I thought it might need to be there (even if there's no real expression). Also, I think D might benefit from being a different size so the height of the plot matches the height of part E. Do you think it's useful to add the 'loadings' on top of the PCA figure?
6. Just a quick note, the sentence "minimal transcriptional and morphological deviations observed suggest that gastruloids grown in these conditions can be used interchangeably with those grown in standard culture, enabling the study of gastrulation in a mechanically controlled environment" is really interesting! This is quite an important finding. Out of curiosity, how low can you go with these gels where it's still a 'gel'?
7. Figure 3 is great. My comments are similar to what I said for Fig. 2 (e.g. PCA plot height, possibly splitting the channel of one of the gastruloids). I think splitting the channel here is quite important, as it's showing where the gene expression is similar/different, and might be a bit hard to see with the size of the figures.
8. Figure 4 is a novel way of showing these changes, and it's interesting to see how much the free-floating gastruloids move (expected though!). For B, how many traces do you think is important to have for this information? I assume multiple replicates? Could the authors discuss how they picked which cells to follow, as there might be differences depending on whether the cells were deep, on the surface, or somewhere in between. For the speed, does this take into account whether the cells are moving in just one plane (the x-y) without much 'up and down' movement (z-plane), or would this sort of cell movement come across as 'slower' since a projection would make it seem like this movement is stationary? Maybe a note in the discussion about this?
9. Final comment on the main manuscript... I can't seem to see any statistical treatment on the effect of stiffness on length or straightness (Fig. 1B, C), or the quantification of normalised intensity (Fig .2B). I'm not sure what you'd do for the latter as by-eye I can see what you're saying, but maybe a measure of difference would be good. Same with Fig. 3B, 4C, S4 (B, D,F), S5D... it would be good to use some statistical treatment here (some sort of anova with multiple comparison adjustments or something else appropriate for the data shape).

Specific comments on M&M

1. Can the authors mention why they use Serum, LIF as well as the 2i components? Surely the whole point of using 2iL is to remove the ambiguity of serum? Considering the absolute pains the authors have gone to removing Matrigel variability, getting good reproducible gastruloids, it just seems strange to have this source of potential future variability. I know many people do this serum+2iL stuff, but I can't lay my hand on who started this or why. It might have been a Lutolf paper using the SBR line ~2020 or something, but a note somewhere would be good about this.
2. By-and-large, the confocal imaging section is nice, but there are some details that are missing which would be good to have. You need to mention whether it's inverted or up-right, the filter sets used (or if it's a variable dichroic, the range of wavelengths that were collected), the lasers (diode?) and the wavelengths. What scan-head was used? I assume the Airyscan was there somewhere?

3. The intensity profiles on the max-projection... I assume the raw data was at the same LUT range between conditions when the max-projection was performed? I think (apologies if I'm wrong, I might be...), the values in the max-projection can change depending on whether the original image has a different display range (for some reason), so making sure they're all the same before-hand is important.

First revision

Author response to reviewers' comments

Reviewer 1:

SUMMARY OF THE ADVANCE MADE IN THIS PAPER AND ITS POTENTIAL SIGNIFICANCE TO THE FIELD

In this manuscript, Pineau and Wong-Ng et. al., present new insights on mechanobiology of mouse gastruloid development. The authors utilize biologically inert dextran hydrogels to study the effect of extracellular stiffness on gastruloid morphogenesis. They report that mouse gastruloids could elongate and establish anteroposterior patterning when embedded in ultra-soft hydrogels and failed elongation in stiffer gels might be due to reduced cell motility. They demonstrate that matrix stiffness does not have a significant influence on the transcriptome of gastruloids and highlight that the timing of gel embedding is critical in inducing changes at the bulk RNA level.

Overall, the authors describe this work as evidence of developmental decoupling of morphogenesis and transcription.

We thank the reviewer for the thorough evaluation of our manuscript and the constructive comments.

SUGGESTIONS TO AUTHORS

1.1. The authors state that "The minimal transcriptional and morphological deviations observed suggest that gastruloids grown in these conditions can be used interchangeably with those grown in standard culture". Although morphology and transcriptome at bulk level seems largely unchanged, tissue organization of control and embedded gastruloids could be different. The authors should support this claim by performing single cell RNA sequencing to better understand the cell type compositions of elongating and non-elongating gastruloids. Moreover, immunostainings or in situ hybridizations to show spatial arrangement of key tissues present in gastruloids (presomitic mesoderm, endoderm, cardiac mesoderm) are necessary.

We thank the reviewer for these important suggestions, and have collected data strengthening the claim that there are limited transcriptional deviations caused by embedding at 96h post seeding. We believe single cell RNA sequencing could be a good idea, however we feel it is out of the scope of this article which is mostly focused on the global effect of changing the mechanical environment on gastruloid development, and would be more interesting in a follow-up work. That said, we have performed deconvolution on our bulkRNAseq data using a previously published single-cell RNA sequencing atlas (Mayran et al., 2023) to estimate the different populations of cell types. **Our analysis revealed that batch-to-batch variability was stronger than the difference between control (non- embedded) gastruloids and those embedded at 96h post seeding. Importantly, no consistent trend was observed in embedded gastruloids, and it further confirmed that the stiffness of the gel does not impact the transcriptional signature. These results are now shown in Figure S5D.**

We also carefully considered the request for staining of other tissues in the gastruloids. The cell line we used for this project behaves similarly to the one described in Veenvliet et al., 2020, and therefore endoderm and cardiac mesoderm are present in a very variable and weak

manner to start with. Knowing that, we decided against staining for these tissues as the results would be very unreliable. However, we performed additional experiments where we stained both SOX2 and FOXC1 (anterior, somitic mesoderm) in control gastruloids and gastruloid embedded at 96h in a 0.7 mM hydrogel. **These images and the corresponding quantifications (antero-posterior intensity profile, boundary position) are now shown in Figure S3 and show that FOXC1 patterning is conserved in embedded gastruloids.**

We hope that these new experiments and analysis that strengthen our claim that embedding at 96h post seeding in ultra-soft hydrogels only has a limited impact on gastruloid development will convince the reviewer.

1.2. As cited in the manuscript, mouse gastruloids embedded in Matrigel elongate efficiently. The authors should include this control condition. It would be interesting to see whether Matrigel embedding promotes straighter gastruloids as in dextran hydrogels.

We thank the reviewer for this very pertinent comment. As highlighted in our manuscript and described in previous studies by other groups (Veenliet et al., 2020, Van den Brink et al., 2020), gastruloids can undergo elongation and form extended structures when embedded in matrigel. In order to compare the effect of embedding in a bioinert, dextran-based gel with those of embedding in matrigel under similar conditions, we performed an experiment in which gastruloids were embedded at 96h post seeding in either 0.7 mM dextran gel, or matrigel diluted to 1mg/mL or 2.5 mg/mL, concentrations corresponding to those used by Veenliet et al., 2020 or Van den Brink et al.,2020. To further assess the role of matrix remodeling, we also tested a condition where matrigel was added to the N2B27 medium (at 1mg/mL) overlaying gastruloids embedded in 0.7 mM dextran gel.

At 120 hours, we observed that gastruloids in 0.7 mM dextran gel had elongated and developed a single pole. In contrast, those embedded in matrigel at either concentration (1 mg/mL or 2.5 mg/mL) consistently exhibited a multipolar morphology, which precluded direct comparison of elongation or straightness between the two embedding conditions. Notably, gastruloids in dextran gel overlaid with matrigel-supplemented N2B27 medium displayed morphologies similar to those embedded in dextran gel alone. This suggests that the observed differences are likely due to the capacity of cells to adhere to and remodel the matrigel matrix rather than an increase in growth factors present in the matrigel. Of note, the goal of this manuscript is not to fully understand the role of the ECM in gastruloid development, but rather to describe how the mechanical component of a bioinert environment impacts patterning and morphogenesis in gastruloid.

We have now added a comment on this point in the main text, as well as images illustrating these observations in Figure S2.

We thank the reviewer for this on point comment. As stated and used in other publications (cite Veenliet paper at least), gastruloids can form extended structures when embedded in matrigel. In order to compare the effect of the dextran based gel to the matrigel, we performed embedding experiment at 96h in the 0.7mM dextran gel, in matrigel diluted to 1mg/mL or 2.5mg/mL. We finally also tested the effect of supplementing matrigel to the supernatant over gastruloids embedded in the dextran based gel at 0.7mM.

The gastruloids in the dextran-based gel display single poles and elongated properly whether there was matrigel in the supernatant. This means that growth factor addition linked to matrigel in the supernatant do not play a significant role at the morphological level. However, when directly embedded in matrigel, gastruloids formed multipolar structures. A proper elongation straightness could not be quantified in these multiple axis bearing gastruloids. Interestingly the gastruloids they formed ultipolar struc as quantifying the elgoation becomes quite pr to have grown in a

Refaire un panel avec qques gastrus choisi a 120h seulement

1.3. The authors compare mechanical properties of dextran hydrogels to Matrigel and show that 1mM dextran hydrogels have lower shear and elastic modulus than 9 mg/ml Matrigel.

However, gastruloids embedded in 1mM dextran hydrogels do not elongate. How would the authors explain this? Can it be due to the rich extracellular matrix protein composition of Matrigel? Is it possible to introduce ECM proteins (laminin, collagen, nidogen) into dextran gels by simply mixing with Matrigel? Would this promote formation of somitic structures? The authors should perform these experiments to better understand the dynamics of axial elongation.

We thank the reviewer for pointing out this seemingly contradictory result. First, we would like to highlight that several important aspects differ between our study and the works referenced by the reviewer. Notably, the cell lines and culture modes used are not the same, and these parameters have been shown to strongly influence both the dynamics of differentiation and the extent of gastruloid development (Blotenburg et al, 2025).

Second, while the mechanical properties of Matrigel and our dextran-based gel can be compared at initial timepoints, their biochemical and physical characteristics over time differ significantly. A key point is that matrigel can be degraded or remodeled by cells secreting matrix metalloproteinases (MMPs), meaning that the initial stiffness might be affected in time, and that cells might degrade the ECM in their vicinity to allow gastruloid elongation.

As described in the previous point, we first ensured that our system (cell line, cell culture method) was capable of supporting elongating gastruloids in Matrigel (**now shown in Figure S2**). To do so, we embedded gastruloids at 96h post seeding and observed clear, elongated multipolar structures by 120h, across both 1mg/mL and 2.5mg/mL Matrigel concentrations. To investigate whether this effect might be due to increased exposure to growth factors or signalling molecules from the matrigel, we supplemented matrigel into the medium overlaying gastruloids embedded in dextran gel. This condition did not result in any notable morphological changes compared to dextran-only embedding, indicating that the ability to elongate in Matrigel likely relies on direct cell-matrix interaction and remodeling rather than growth factor content.

Supporting this, our bulk RNA-sequencing data reveal that cells in gastruloids express several MMPs (in particular *Mmp27*, *Mmp13*, *Mmp12* and *Mmp10*), even more so in embedded gastruloids, as shown in Figure S5E. This suggests active remodeling of Matrigel in the vicinity of the organoid, potentially lowering the local elastic modulus and enabling elongation. In contrast; the bioinert dextran gel resists such remodeling, which likely restricts elongation despite its lower initial stiffness.

The reviewer's suggestion to incorporate ECM proteins into the dextran gel is indeed an interesting avenue and aligns with ongoing efforts in hydrogel bioengineering. However, development of such hybrid gels and systematically exploring their properties represents a broader project and is outside the scope of the current study.

Likewise, we would like to highlight that studying the dynamics of axial elongation is not the primary focus of this work. Our aim here is to explore how a defined, bioinert environment influences patterning and morphogenesis in gastruloids, and to propose a new tool that allows the community to access what happens during gastruloid elongation and polarity axis establishment.

1.4. The authors offer gel embedding as a better set up for live imaging, however they do not provide timelapse videos of elongating gastruloids. Providing such videos would strengthen this claim.

We thank the reviewer for pointing this out. Indeed, we had forgotten to add videos illustrating gastruloid elongation in gels. **We have now added time lapse movies acquired on a videomicroscope** to support the fact that ultra-low stiffness embedding in these bioinert gels gives access to long term imaging of gastruloid development.

1.5. What is the mesh size of the ultra-soft and stiff gels? Can it be possible that in addition to physical constraints, stiffer gels also entrap secreted molecules that might have influence on axial elongation?

We thank the reviewer for this insightful comment. We use dextran-based hydrogels from Cellendes. These matrices are fully synthetic, and allow to tune mechanical properties by adjusting functional group concentration. Although the precise mesh size isn't provided by the manufacturer, estimates can be derived from the Elastic modulus and temperature of the experiment, using elastic blob theory (Tsuji et al., Gels, 2018). Considering an experimental setup at 37°C, and the gels used in our study: between 0.7 mM (3 Pa) and 1.5 mM (300 Pa), the resulting mesh sizes would be around 110 nm for the softest gels, down to 24 nm for the stiffest gels (see also <https://www.cellendes.com/technology/mesh-size-of-hydrogels>).

It is estimated that nutrients, morphogens, growth factors or cytokines have hydrodynamic radii in the range of 5nm (Armstrong et al., Biophys J., 2004), smaller than the mesh size of the stiffest hydrogels used. Therefore, the organoid should not be deprived of these important actors coming from the media, and secreted signaling molecules should be able to diffuse away from the organoid. However, according to the existing literature, the coefficient of diffusion of a particle within a gel is affected by mesh size. In particular, the diffusion coefficient D of a particle of diameter d within a gel of mesh size ξ would vary as:

$$D = \frac{\xi^2}{\tau_0 N_x^2 \phi} \left(\frac{b N_x^{1/2}}{d} \right) e^{-d^2 / N_x b^2} \quad (\text{from Moncure et al., The Journal of Physical Chemistry B, 2022})$$

With:

τ_0 : monomer relaxation time ; N_x : number of monomer units in between cross-links ; ϕ : Polymer volume fraction ; b : Kuhn monomer length

With that model, the diffusion coefficient varies as the square of the mesh size of the gel, resulting in about a 20-fold decrease in diffusion coefficient between the ultra-soft gels (0.7mM, ~3 Pa, ~110nm mesh size) and the stiffest ones (1.5 mM, ~300 Pa, ~24nm mesh size). The difference in diffusion coefficient between the two stiffest gels (1.0 mM, ~30 Pa, ~54nm mesh size and 1.5 mM, ~300 Pa, ~24nm mesh size) where strong differences are observed in term of A-P axis formation, would be only 4-fold.

We acknowledge that this could also be a factor affecting gastruloid development in embedded systems, especially in the case of the stiffest hydrogel. However, given our results showing that gel concentration does not strongly affect the transcriptional profile nor the cell type composition (see bulk RNA sequencing results, in particular deconvolution in Figure S5D), we believe that a potential effect could be rather on directed cell migration than on cell differentiation. In addition, when considering diffusion of secreted factors, those would be strongly diluted in control, non-embedded culture conditions, and their potential effect would likely mostly rely on those trapped and diffusing within the gastruloid. While gel embedding would change the diffusion coefficient in the surrounding environment, its impact on diffusion within the gastruloid, at least in the early time points when the constraints on the gastruloid are very small, would be negligible. Finally, the quantitative impact of diffusion coefficient within an organism on the patterning scales might not be relevant (Gregor et al. 2005). **We have now added a part discussing gel mesh size and diffusion in the discussion.**

1.6 A major drawback of conventional mouse gastruloid culture is the bending and loss of elongated morphology at the later stages of culture. It is particularly interesting to see dextran embedded gastruloids adopt a straighter morphology than controls. If the hydrogels are stable, would these "straighter" gastruloids maintain the elongated morphology up to 168h? If not, can they be freed of the hydrogel at 120h and maintained in suspension longer?

We thank the reviewer for suggesting this experiment, that could bring an additional advantage to our embedding approach. **We performed an experiment embedding gastruloids at 96h in a 0.7mM dextran gel, and keeping them in culture until 120h, 144h and 168h (now shown in Figure S2).** We could not conclude that gel embedding prevents the collapse of the gastruloid. Cells accumulate around the organoid in time and prevent a good morphological assessment. However, we believe our gel embedding system does not allow to maintain the gastruloid morphology for longer periods than in liquid culture.

Because dextranase specifically degrades the Dextran gel backbone. There should be no issue in

removing the gastruloids to try maintaining them in suspension longer. However, we did not assess the impact of gel embedding on further disembedded growth/development as we do not believe this to be within the scope of our study.

1.7 Early embedding seems to prevent elongation. Would embedding at later timepoints (120h) increase longevity of gastruloids?

We thank the reviewer for seeing possible new applications for our embedding system. As described in the previous point and shown in **Figure S2**, we found that embedding in a dextran gel at 96h did not prevent organoid collapse after 120h. Given these observations, we do not believe that embedding at 120h would help maintain the morphology of the organoid. In contrast, we observed that embedding in Matrigel at 96h, while it resulted in a multipolar morphology at 120h in our hands, allowed us to obtain a very elongated and non-collapsed morphology at 144h post seeding. This shows that longevity can be increased but it is highly dependent on the nature of the gel.

1.8 Increased cellular compression and decreased cell proliferation could be quantified by YAP and Ki67(or EdU) stainings in gastruloids embedded at 72h in 1.5 mM gels.

We thank the reviewer for these suggestions. We believe YAP nuclear translocation would be very hard to analyze in these organoids, as it would necessitate separating nuclear from cytoplasmic areas, in a 3D, dense environment. **We tried to assess cell density by looking at the nuclear signal (DAPI), and measure density of dividing cells by staining for H3S10P, a cell division marker (Figure S8A).**

To estimate cell density, as we were not able to segment nuclei, we measured the average intensity of the DAPI signal in similar conditions (same depth within the organoid), but did not find a significant difference (Figure S8B). We also measured the density of detected H3S10P^{high} cells, and did not find a significant difference (Figure S8C), and added a comment on this result in the main text. Of note, while this suggests that there is no significant impairment of cell division at 120h post seeding in 1.5mM gels as compared to 1.0mM gels, there could have been differences in cell division between 72h and 120h, while constraints on the organoid were still accumulating, and the state at 120h might be an equilibrium state where both gastruloids in both conditions have reached their limits.

1.9 In discussion, the text reads "Gastruloids embedded in stiffer hydrogels (1.5mM) exhibited significantly reduced cell motility compared to those in softer hydrogels (1.0 mM)" however the related Fig. 4C does not report any significance. Authors should either replicate the experiment to test significance or modify the text. Overall, it would be more relevant to plot all replicates of morphological quantifications (length, straightness ratio) in the same graphs and report statistical significance. This is missing throughout the paper. Supp. Fig. 5D attempts to do this but misses statistical analysis.

We thank the reviewer for this comment and have tried to answer it throughout the paper. **We have now added statistical analysis in all graphs for individual replicates, and added for each measure a graph showing all replicates (in supplementary figures).** We chose not to perform statistical analysis on pooled data, as gastruloids are an experimental system that tends to have substantial batch-to-batch variability, and we did not want this to affect data interpretation. **After statistical analysis, the gastruloids embedded in stiffer hydrogels (1.5mM) do exhibit significantly reduced cell motility as compared to those in softer hydrogels (1.0mM).**

1.10 Supp. Fig. 7a: What do the colors indicate? Is it the depth of the location the cell is found? It is difficult to judge without a legend.

We thank the reviewer for pointing out this legend that we had forgotten. Colors indicate the time points when the segment of the trajectory has been recorded. **This information has now been added to the figure caption and a color bar was added on the Figure (now Figure S10).**

1.11Supp. Fig. 7b: Do different colors indicate individual cells? Which cells are BRA+ and which are H2B+? It would help a lot if a mask of gastruloid is also displayed, similar to S7a.

We thank the reviewer for highlighting this point that needed clarification. Here, each color corresponds to a cell trajectory. All cells tracked are H2B-iRFP+, as it is with this channel that we performed the tracking, and we did not cross this information with the Tprom-mVenus signal, as this second channel was used only to make sure we did conserve the phenotype (BRA+ pole formation or not). **We adapted the legend (now Figure S10) to clarify what is plotted:** this panel represents tracks of cells from several organoids, where the tracks have been centered to all start at the coordinate (0,0), and therefore are not within the frame of reference of the gastruloid. For that reason, we cannot add the outline of the gastruloid. This is only a representation we found useful to visualize how much cells explore the environment in 2D.

Reviewer 2:

In this work Pineau et al. explore the effect of hydrogel embedding on the formation of mouse 3D gastruloids. To do this, the authors use dextran-based PEG hydrogels with varying stiffnesses. When gastruloids are embedded in ultra- soft hydrogels (<30 Pa), they are able to polarize and elongate normally as in control non-embedded gastruloids. However, when the stiffness is increased (>30 Pa), embedding at 96h impairs elongation while leaving polarization unaffected. In contrast, if hydrogel embedding is done at 72h, for very stiff gels (~300 Pa), both polarization and elongation are impaired. Finally, in this last condition, the authors track cell movement and report impaired cell motility.

The work is sound and the manuscript is well written. In particular, I believe the PEG hydrogel embedding method is of special interest to the gastruloid community. Coincidentally, the fact that embedding in ultra-soft hydrogels improves cell tracking without affecting gastruloid development is also an interesting result of the paper. Below I enumerate a list of major and minor comments which I think can significantly improve the manuscript. Provided that my comments are successfully addressed, I am happy to recommend the manuscript for publication in Development.

We thank the reviewer for the positive comments on the manuscript and hope to answer all of the points to fully convince the referee.

Major comments

1) The title of the manuscript states that "fine-tuning mechanical constraints uncouples patterning and gene expression". I do not understand how the authors reach this conclusion from the obtained results. In particular, in their discussion they say that "[...] the transcriptional program proceeds autonomously within each cell, relying on local or short-range cues rather than long-range gradients. These findings challenge conventional views of tight coordination between transcription and patterning". I believe the message the authors are trying to convey is that their results challenge a reaction-diffusion mechanism (long-range cues) and support a cell sorting mechanism (short-range cues) for symmetry breaking. However, I find the title misleading, suggesting that mechanical embedding actively uncouples patterning and gene expression, and thus when the system is not embedded patterning and gene expression are coupled.

We thank the reviewer for raising this point and apologize for the misleading structure of our title. We did not mean to suggest that the act of embedding actively uncouples patterning and expression programs, but rather they are uncoupled and by playing with the constraints it can be revealed as such. We prefer not to put too much emphasis on the mechanisms that lead to such decoupling as we don't have much evidence for either. **We have modified our title to "Fine-tuning mechanical constraints reveals uncoupled patterning and gene expression programs in murine pseudo-embryos" and hope this makes our message clearer.**

2) The hydrogel stiffness increases logarithmically in the mM range of linking functions as shown in Fig. S1C,D. As reported in the paper, the major effect on embedding is found around ~30-300 Pa (1 and 1.5 mM), however this is an order of magnitude in range. The uncertainty in stiffness for the 1 mM conditions seems to be 30 ± 10 Pa from Fig. S1C. Given that the dispersion in gel stiffnesses is of ~10 Pa, my question is why the authors did not explore in more detail the range of 10-100 Pa. Indeed, recent work has reported that the stiffness of gastruloids is ~30 Pa (see Oriola et al. bioRxiv 2024) and thus this would be the interesting range to study rather than very large stiffnesses (300 Pa).

We thank the reviewer for highlighting this, and appreciate their thinking. We operated with a linear range of concentration, which translates into a non-linear range of Elastic Moduli. In our hands, given the non-linearity of gel stiffness relative to concentration, and the difficulty to change concentration more finely than we did, it would be hard to do a systematic study looking into stiffnesses between 10 and 100 Pa. In particular, if we were to attempt at pinpointing at which modulus elongation or polarization is stopped, we would need to explore more finely values between 2 Pa (0.8mM) and 30 Pa (1.0 mM), as at 30 Pa elongation is already prevented, making the study between 30 Pa and 100 Pa less pertinent, especially since we see very little differences in terms of morphology between 30 Pa (1.0 mM) and 300 Pa (1.5 mM). **We have, however, added a comment in the discussion regarding the stiffness of gastruloids reported by Oriola et al., Biorxiv 2024, as we believe it is interesting as a point of comparison to the elastic moduli used in our study.**

3) In Fig 4C, the authors quantify what they call "cell migration speed" and find a ~30% decrease in motility from 1 mM to 1.5 mM hydrogels. Taking into account this is an order of magnitude change in stiffness (from 30 to 300 Pa), I find the change in cell motility not very dramatic. Indeed, one could expect that the mean cell motility is not greatly affected but the diffusion coefficient is due to caging effects. Therefore, I think it would be instructive to also compute the relative mean squared displacement to explore the type of anomalous diffusion found in both conditions, which might be more meaningful than the mean cell speed.

We thank the referee for suggesting to find other ways of quantifying the difference in migration between gastruloids embedded at 72h in 1.0mM or 1.5 mM gels. First, the decrease in mean cell migration speed, while not dramatic, is statistically significant (**we have now added statistical analysis in the new panel Figure 4D**) and not negligible. In addition, we have now generated a new graph representing all cell trajectories of cells in a 1.0mM or 1.5mM gel, centered to start at the same origin point, which helps visualizing that cells in organoids embedded in a 1.0mM gel appear to be migrating further than those in a stiffer gel (**Figure 4C**).

We have performed other types of analysis, computing the confinement ratio (net distance travelled/total distance travelled), the mean directional change rate, and the MSD, now all shown in Figure S10C-G. We could not detect any strong effect on confinement ratio or mean directional change rate. Below is also the MSD plot in log-log scale, the diffusion coefficient A as well as the diffusion exponent α , both extracted from the computation of MSD. While looking at the MSD plot gives the impression that trajectories in the 1mM condition could have higher diffusion coefficient and/or exponent, the quantification of those values does not confirm this. Interestingly, in both conditions, the diffusion exponent α is, on average, above one, suggesting a super-diffusive behavior. However, we believe that the measure of MSD in our case is very approximate, as trajectories are relatively short (10-25 frames) and frame rate (3 image/hour) might not be high enough.

Minor comments

1) In order to improve the readability of the paper I would suggest to include Fig. S1 as Figure 1 on the main text. I think this is necessary for the reader to clarify details on the method and the conditions used. In addition, panels Fig. S1C and D I think are really important and should be in the main text.

We thank the reviewer for this comment that will help the clarity of the paper. **We have now combined the previous Figure 1 with Figure S1 panel A, C, D, E in the main text.**

2) In Fig. 1 it is shown that ultra-soft hydrogels (1-5 Pa) are sufficient to straighten gastruloids however only when the stiffness is around ~30 Pa elongation is impaired. Isn't this result somehow implying that gastruloids are easier to bend than to stretch? This would be an interesting result to discuss.

We thank the reviewer for this insightful comment highlighting a surprising result. Our findings indeed suggest that gastruloids are more easily bent than stretched. Ultra-soft hydrogels (1-5 Pa) are sufficient to straighten gastruloids, while axial elongation is only significantly impaired at higher stiffness levels (~30 Pa). This likely reflects a fundamental mechanical asymmetry: bending a structure generally requires less force than stretching it, as described in classical beam theory, where bending resistance scales with the material's flexural rigidity, while axial resistance scales linearly with tensile stiffness.

This mechanical asymmetry—where bending resistance is much lower than tensile resistance—has been shown in both biological and biomimetic fibrous networks (collagen, matrigel, see Prince et al., PNAS, 2023), where the bending modulus of individual fibers is orders of magnitude smaller than their stretching modulus, resulting in an anisotropic behavior.

While this is an interesting point, the mechanical behavior of gastruloids is not well known. Hence, we do not feel confident enough to address this specific topic in the discussion of our manuscript.

3) The authors suggest that the inability of gastruloids to establish a BRA/SOX2 pole when embedded at 72h in 1.5 mM hydrogels is due to impaired cell motility and not due to transcriptional changes. They reason that this is likely due to an increase in cell density which is also supported by the fact that the aggregates are smaller in this condition. This would point towards a cell sorting mechanism for pole formation which is in line with recent work (McNamara et al. NCB 2024, Gsell et al. 2025, Oriola et al. bioRxiv 2024). It would be interesting to add this in the discussion.

We thank the reviewer for this interesting suggestion to enrich the discussion. We have attempted to estimate changes in cell density in our system (see answer to reviewer 1, 1.8, and Figure S8) by comparing DAPI intensity, but our measure remained inconclusive. Still, it is possible that gastruloids embedded in stiffer gels are more dense, leading to impaired ability of cells to sort and form the different regions. This is also in line with the work presented in Mayran et al., 2023, as mentioned in the previous version of the discussion. **We now mention defects in cell sorting mechanisms as a potential reason for the observed defect in posterior pole formation in the new version of the discussion, supported by the suggested references.**

4) From the methods section I understand that cells are cultured in 2i LIF conditions. I would include this explicitly in Fig. S1 in panel A.

We have now added this information in the panel, which has been moved to Figure 1.

Reviewer 3:

SUMMARY OF THE ADVANCE MADE IN THIS PAPER AND ITS POTENTIAL SIGNIFICANCE TO THE FIELD

Pineau et al are asking a fundamental question: how does the balance between mechanical and genetic programmes influence morphogenesis and cell fate? To answer this, they couple the Gastruloid model system with hydrogels that have been tuned to specific stiffness. Critically this approach is a good diversion from the typical Matrigel approaches which, although providing different stiffnesses, can't be tuned and aren't inert. They find that whereas low stiffness hydrogels can produce the typical elongations and gene expression patterns, stiffer hydrogels maintain expression but not polarisation. In addition, the

timing of hydrogel embedding is also important, altering the transcriptional profiling. Live imaging approaches add a lovely dimension to this work, and allows a mechanistic understanding of these differences, suggesting that cell mobility is a critical input into these processes.

As a quick summary, I think this manuscript is exactly what the gastruloid system should be used for: uncoupling the different inputs that cells and embryos would receive during development in a proper quantitative and methodological manner, understanding what input regulates what process. The data are clear, and the conclusions match what is presented, although it would be good to see statistics used in the quantification of gastruloid metrics (see below). The work is a very important addition to the field, really show-cases how useful the system is, and was a pleasure to read. I only have minor comments.

We thank the referee for the support and enthusiastic comments.

SUGGESTIONS TO AUTHORS

Specific comments:

1. For Fig. 1, I like the measure of gastruloid straightness; as far as I can remember, this hasn't been done before, even though many of us in the field mention the different morphologies in passing. Do the authors think elongation index could also be used here? I think it might add an additional dimension which is straightforward to do.

We thank the referee for suggesting adding the Elongation index that will help describe gastruloid morphology, and we are happy the new metric we created to measure straightness of gastruloids is useful for the community.

Elongation Index, as described in M. U. Girgin et al. 2021, has now been added in Figure 1 and Figure S1 and helps describe better the effects on gastruloid morphology. However, in our hands, the effects on the value of elongation index were quite variable, with the only robust effect being the very limited elongation of gastruloids in 1.0 mM gels.

2. In addition to Fig. 1, it might be good not only to say the N, but also the number of replicate experiments in the figure for each time-point (only in the legend, not the figure); it just adds more weight to the power of gastruloids in getting replicate experiments compared with animals. I know you show in the supplemental the three replicates, but a quick note of this in the legend is fine.

We agree with the reviewer that this will help highlight one of the advantages of using gastruloids, which is the possibility of obtaining replicate experiments in a more controlled manner than with animal experimentation. **We have now added in the legend of Figure 1 the number of replicate experiments, and where to find the other replicate experiments.**

3. On page 4, the gene names are in capitals... I think this ought to be changes to lower-case with the exception of the first letter.

We have now checked carefully throughout the text to make sure that when we refer to proteins, names are fully capitalized, and when we refer to genes, only the first letter is capitalized and the name is in italics.

4. The sentence "fluorescence intensities were normalized for both intensity and length, enabling a comparison of the spatial expression profiles..." is interesting, important, and probably should be an echo what's in the M&M section just so we don't have to hunt through the manuscript to find what this is normalisation is

We have added a sentence describing the normalization procedure in the main text.

5. Figure 2: This is a lovely figure, but two things I think need modifying slightly. It might be useful if (either here or supplemental) the authors show maybe one gastruloid with the colours split so you can see where they are? I really like seeing the three replicate stains, but it could be useful to see the different channels somewhere. In B and C, it would be good if the legend spanned the two graphs so it's easier to see that it belongs to both; also I had to check and see whether you included the no chi quantification. Should this be in here, as it's coloured 'red'

and I thought it might need to be there (even if there's no real expression). Also, I think D might benefit from being a different size so the height of the plot matches the height of part E. Do you think it's useful to add the 'loadings' on top of the PCA figure?

We thank the reviewer for helping with harmony and readability of figures. **We have adapted Figure 2 according to these suggestions: we added a gastruloid with split colours, and made Figure 2E shorter so as to match the size of the PCA plot**, and hope this new version will help the readability of the figure. We tried to add the loadings on top of the PCA plot (see below), but feel that it is not easy to read and will not be helping clarity of the figure. We chose to, instead, add a graph with only the loadings in Supplementary Figure S5.

NOTE: We have removed unpublished data that had been provided for the referees in confidence.

6. Just a quick note, the sentence "minimal transcriptional and morphological deviations observed suggest that gastruloids grown in these conditions can be used interchangeably with those grown in standard culture, enabling the study of gastrulation in a mechanically controlled environment" is really interesting! This is quite an important finding. Out of curiosity, how low can you go with these gels where it's still a 'gel'?

A gel is defined as a cross-linked polymer network diluted and swollen in a liquid phase. A classical definition of gelation resides in the intersection of the storage modulus (G') and the loss modulus (G''). Crossing the gelation point means we have a gel in this classical sense. However, in our case, we could not see the gelation point as our gels were too diluted. Instead, we can define our matrix as a gel as long as the different components (in this case, Dextran and PEG through the maleimide and thiol reactive functions) are cross-linked within the liquid phase. Although an absolute value of function concentration below which this crosslinking cannot happen anymore, we are confident it is still the case for the lowest concentration presented here. Indeed, when measuring the kinetics of shear modulus evolution, we can see that this modulus is evolving in time, indicating a crosslinking reaction, as shown in the graph below (which is a zoom in from the graph now presented in Figure S1A). Another indication we still have a gel at our lowest concentrations is the fact that the gastruloids barely move during a 24 hour movie (Fig 4A) while unembedded ones do. This means that on our acquisition timescale, the matrix behaves roughly like a solid or a gel.

NOTE: We have removed unpublished data that had been provided for the referees in confidence.

7. Figure 3 is great. My comments are similar to what I said for Fig. 2 (e.g. PCA plot height, possibly splitting the channel of one of the gastruloids). I think splitting the channel here is quite important, as it's showing where the gene expression is similar/different, and might be a bit hard to see with the size of the figures.

We thank the reviewer for helping with figure design, and **have adapted the figure by adding split color images for one gastruloid per condition, and changing the height of the Differentially Expressed genes plot to match the one of the PCA**. We have made the choice of matching the aspect ratio of the PCA plot to the relative contributions to the variance as we find it helps highlighting the extent of differences between samples, and hope that the referee will find the new version of Figure 3 more harmonious and readable. **We also added the loadings of the PCA in Figure S9.**

8. Figure 4 is a novel way of showing these changes, and it's interesting to see how much the free-floating gastruloids move (expected though!). For B, how many traces do you think is important to have for this information? I assume multiple replicates? Could the authors discuss how they picked which cells to follow, as there might be differences depending on whether the cells were deep, on the surface, or somewhere in between. For the speed, does this take into account whether the cells are moving in just one plane (the x-y) without much 'up and down' movement (z-plane), or would this sort of cell movement come across as 'slower' since a projection would make it seem like this movement is stationary? **Maybe a note in the discussion about this?**

“For B, how many traces do you think is important to have for this information? I assume multiple replicates?”

In Figure 4B, we show an example with a subset of traces within a single organoid, which is of course not sufficient to conclude anything. However, having performed tracking in two organoids per condition, we extracted the mean speed for each track in the organoids, with tracks coming from different organoids represented with different shapes (triangle or circle). Since the triangle and circle symbols appear well mixed, it suggests that there is very little inter-gastruloid variation in cell migration speed within the same experiment. Given this observation, we believe very few gastruloids would be needed to reach a conclusion for a replicate. Of course, this does not exempt from performing replicate experiments.

“Could the authors discuss how they picked which cells to follow, as there might be differences depending on whether the cells were deep, on the surface, or somewhere in between.”

Cells to follow were selected in a stack of 38 slices around the midplane of the organoid, and were mostly near the surface: not at the surface to avoid some bright cell debris that can accumulate there, and not in the core as imaging quality and contrast was too degraded due to optical limitations. We are aware that there might be differences from what would be observed in cells at the center of the organoid, or really at the surface, but as we were not able to properly track those, we could not compare. However, since cell tracking was performed in the same way in both conditions, the comparison between the two can still be done.

“For the speed, does this take into account whether the cells are moving in just one plane (the x-y) without much 'up and down' movement (z-plane), or would this sort of cell movement come across as 'slower' since a projection would make it seem like this movement is stationary? Maybe a note in the discussion about this?”

Thanks to the large dataset obtained with Light-Sheet imaging, cells were tracked in 3D in a z-stack around the midplane of the gastruloid, so movement in z was accounted for in all directions when measuring the speed. However, this movement is not represented in the 2D trajectories plotted as illustrations in Figure 4B and S8B.

9. Final comment on the main manuscript... I can't seem to see any statistical treatment on the effect of stiffness on length or straightness (Fig. 1B, C), or the quantification of normalised intensity (Fig. 2B). I'm not sure what you'd do for the latter as by-eye I can see what you're saying, but maybe a measure of difference would be good. Same with Fig. 3B, 4C, S4 (B, D, F), S5D... it would be good to use some statistical treatment here (some sort of anova with multiple comparison adjustments or something else appropriate for the data shape).

We thank the reviewer for this comment that helps improve the rigorousness of this article. **We have now included statistical tests for the effects on length, straightness and elongation index (Kruskal-Wallis with a correction for multiple comparison) as well as cell speed (Mann-Whitney) in this version.**

Regarding the quantification of the differences in profiles, **we have defined a value for the boundary position of BRA, SOX2 and FOXC1 profiles, as described in Bennabi et al., 2024, and included this measurement with the appropriate statistical tests in Figure S3 and S4.**

Specific comments on M&M

1. Can the authors mention why they use Serum, LIF as well as the 2i components? Surely the whole point of using 2iL is to remove the ambiguity of serum? Considering the absolute pains the authors have gone to removing Matrigel variability, getting good reproducible gastruloids, it just seems strange to have this source of potential future variability. I know many people do this serum+2iL stuff, but I can't lay my hand on who started this or why. It might have been a Lutolf paper using the SBR line ~2020 or something, but a note somewhere would be good about this.

The reviewer is correct that serum could constitute a source of variation. However, generally speaking by using ES grade serum in Serum LIF culture condition, the serum would not generally be the major source of variability, but rather that the absence of 2i in the LIF Serum culture condition creates ES cell heterogeneity. This leads to a more variable morphology of gastruloids as the starting cell state is heterogeneous. In contrast, by using 2i, the starting cell population is more homogeneous and this leads to more robust gastruloids (Beccari et al. 2018, vs Merle et al. 2024). Finally, long term culture of ES cells in N2B27 with 2i LIF, leads to colonies which can easily detach. By including the Serum (and thus using DMEM), the ES colonies remain firmly attached to the plate.

We have now added a comment about this point in the main text, as well as highlighted the information about cells being cultured in Serum+2iL media in the new Figure 1, that includes the gastruloid generation and embedding protocol previously shown in Figure S1.

2. By-and-large, the confocal imaging section is nice, but there are some details that are missing which would be good to have. You need to mention whether it's inverted or up-right, the filter sets used (or if it's a variable dichroic, the range of wavelengths that were collected), the lasers (diode?) and the wavelengths. What scan-head was used? I assume the Airyscan was there somewhere?

We thank the referee for ensuring that our MMs would be thorough and detailed. **We have now added this information in the manuscript.** Of note, Airyscan was not used for these acquisitions.

3. The intensity profiles on the max-projection... I assume the raw data was at the same LUT range between conditions when the max-projection was performed? I think (apologies if I'm wrong, I might be...), the values in the max-projection can change depending on whether the original image has a different display range (for some reason), so making sure they're all the same before-hand is important.

We understand the reviewer's concern and thank the reviewer for this thorough evaluation. In our analysis, we have verified that the max projections were done on the actual raw values in Python, not on displayed values, to ensure no bias linked to display settings, although as far as we know, there is no such effect of display on imageJ or Python.

Second decision letter

MS ID#: dev.204711R1

MS TITLE: Fine-tuning mechanical constraints reveals uncoupled patterning and gene expression in murine pseudo-embryos

AUTHORS: Judith Pineau, Jerome Wong-Ng, Alexandre Mayran, Lucille Lopez-Delisle, Pierre Osteil, Armin Shoushtarizadeh, Denis Duboule, Samy Gobaa and Thomas Gregor

Dear Dr Gregor,

I am happy to tell you that your manuscript has been accepted for publication in Development, pending our standard publication integrity checks.

Reviewer 2

Advance summary and potential significance to field

The authors successfully addressed all my comments and suggestions and I believe the manuscript is now ready for publication.

I would only like to make a last comment regarding the MSD analysis. I acknowledge the fact that the authors tried to extract the diffusion coefficient from the time-lapse movies as I suggested. However, as the authors mention in point 3 of the rebuttal letter, I agree that a frame rate of 3 images/h is most likely too slow to capture the diffusive motion. I believe this is reflected in Fig. S10G, where the diffusion coefficients seem to be virtually zero (I could not find the diffusion coefficient values in the manuscript). Therefore, I would suggest to remove panel G from Fig. S10.

Reviewer 3

Advance summary and potential significance to field

As I mentioned in my first review, this is an excellent paper, highly significant and an important addition to the field. I'd like to thank the authors for their response to my comments and I feel they have addressed my (and the other reviewers') comments appropriately.

I have no other comments on this manuscript; happy to see it published!